# Outlier-robust estimation of a sparse linear model using $\ell_1$-penalized Huber's $M$-estimator

**Arnak S. Dalalyan**
ENSAE Paristech-CREST
arnak.dalalyan@ensae.fr

**Philip Thompson**
ENSAE Paristech-CREST
philipthomp@gmail.com

## Abstract

We study the problem of estimating a $p$-dimensional $s$-sparse vector in a linear model with Gaussian design and additive noise. In the case where the labels are contaminated by at most $o$ adversarial outliers, we prove that the $\ell_1$-penalized Huber's $M$-estimator based on $n$ samples attains the optimal rate of convergence $(s/n)^{1/2} + (o/n)$, up to a logarithmic factor. For more general design matrices, our results highlight the importance of two properties: the transfer principle and the incoherence property. These properties with suitable constants are shown to yield the optimal rates, up to log-factors, of robust estimation with adversarial contamination.

## 1   Introduction

Is it possible to attain optimal rates of estimation in outlier-robust sparse regression using penalized empirical risk minimization (PERM) with convex loss and convex penalties? Current state of literature on robust estimation does not answer this question. Furthermore, it contains some signals that might suggest that the answer to this question is negative. First, it has been shown in (Chen et al., 2013, Theorem 1) that in the case of adversarially corrupted samples, no method based on penalized empirical loss minimization, with convex loss and convex penalty, can lead to consistent support recovery. The authors then advocate for robustifying the $\ell_1$-penalized least-squares estimators by replacing usual scalar products by their trimmed counterparts. Second, (Chen et al., 2018) established that in the multivariate Gaussian model subject to Huber's contamination, coordinatewise median—which is the ERM for the $\ell_1$-loss—is sub-optimal. Similar result was proved in (Lai et al., 2016, Prop. 2.1) for the geometric median, the ERM corresponding to the $\ell_2$-loss. These negative results prompted researchers to use other techniques, often of higher computational complexity, to solve the problem of outlier-corrupted sparse linear regression.

In the present work, we prove that the $\ell_1$-penalized empirical risk minimizer based on Huber's loss is minimax-rate-optimal, up to possible logarithmic factors. Naturally, this result is not valid in the most general situation, but we demonstrate its validity under the assumptions that the design matrix satisfies some incoherence condition and only the response is subject to contamination. The incoherence condition is shown to be satisfied by the Gaussian design with a covariance matrix that has bounded and bounded away from zero diagonal entries. This relatively simple setting is chosen in order to convey the main message of this work: *for properly chosen convex loss and convex penalty functions, the PERM is minimax-rate-optimal in sparse linear regression with adversarially corrupted labels.*

To describe more precisely the aforementioned optimality result, let $\mathcal{D}_n^\circ = \{(\boldsymbol{X}_i, y_i^\circ); i = 1, \ldots, n\}$ be iid feature-label pairs such that $\boldsymbol{X}_i \in \mathbb{R}^p$ are Gaussian with zero mean and covariance matrix $\boldsymbol{\Sigma}$ and $y_i^\circ$ are defined by the linear model

$$y_i^\circ = \boldsymbol{X}_i^\top \boldsymbol{\beta}^* + \xi_i, \qquad i = 1, \ldots, n,$$

where the random noise $\xi_i$, independent of $\boldsymbol{X}_i$, is Gaussian with zero mean and variance $\sigma^2$. Instead of observing the "clean" data $\mathcal{D}_n^\circ$, we have access to a contaminated version of it, $\mathcal{D}_n = \{(\boldsymbol{X}_i, y_i); i = 1, \ldots, n\}$, in which a small number $o \in \{1, \ldots, n\}$ of labels $y_i^\circ$ are replaced by an arbitrary value. Setting $\theta_i^* = (y_i - y_i^\circ)/\sqrt{n}$, and using the matrix-vector notation, the described model can be written as

$$\boldsymbol{Y} = \mathbf{X}\boldsymbol{\beta}^* + \sqrt{n}\,\boldsymbol{\theta}^* + \boldsymbol{\xi}, \tag{1}$$

where $\mathbf{X} = [\boldsymbol{X}_1^\top; \ldots; \boldsymbol{X}_n^\top]$ is the $n \times p$ design matrix, $\boldsymbol{Y} = (y_1, \ldots, y_n)^\top$ is the response vector, $\boldsymbol{\theta}^* = (\theta_1^*, \ldots, \theta_n^*)^\top$ is the contamination and $\boldsymbol{\xi} = (\xi_1, \ldots, \xi_n)^\top$ is the noise vector. The goal is to estimate the vector $\boldsymbol{\beta}^* \in \mathbb{R}^p$. The dimension $p$ is assumed to be large, possibly larger than $n$ but, for some small value $s \in \{1, \ldots, p\}$, the vector $\boldsymbol{\beta}^*$ is assumed to be $s$-sparse: $\|\boldsymbol{\beta}^*\|_0 = \text{Card}\{j : \beta^* \neq 0\} \leq s$. In such a setting, it is well-known that if we have access to the clean data $\mathcal{D}_n^\circ$ and measure the quality of an estimator $\widehat{\boldsymbol{\beta}}$ by the Mahalanobis norm[1] $\|\boldsymbol{\Sigma}^{1/2}(\widehat{\boldsymbol{\beta}} - \boldsymbol{\beta}^*)\|_2$, the optimal rate is

$$r^\circ(n, p, s) = \sigma\Big(\frac{s\log(p/s)}{n}\Big)^{1/2}.$$

In the outlier-contaminated setting, *i.e.*, when $\mathcal{D}_n^\circ$ is unavailable but one has access to $\mathcal{D}_n$, the minimax-optimal-rate (Chen et al., 2016) takes the form

$$r(n, p, s, o) = \sigma\Big(\frac{s\log(p/s)}{n}\Big)^{1/2} + \frac{\sigma o}{n}. \tag{2}$$

The first estimators proved to attain this rate (Chen et al., 2016; Gao, 2017) were computationally intractable[2] for large $p$, $s$ and $o$. This motivated several authors to search for polynomial-time algorithms attaining nearly optimal rate; the most relevant results will be reviewed later in this work.

The assumption that only a small number $o$ of labels are contaminated by outliers implies that the vector $\boldsymbol{\theta}^*$ in (1) is $o$-sparse. In order to take advantage of sparsity of both $\boldsymbol{\beta}^*$ and $\boldsymbol{\theta}^*$ while ensuring computational tractability of the resulting estimator, a natural approach studied in several papers (Laska et al., 2009; Nguyen and Tran, 2013; Dalalyan and Chen, 2012) is to use some version of the $\ell_1$-penalized ERM. This corresponds to defining

$$\widehat{\boldsymbol{\beta}} \in \arg\min_{\boldsymbol{\beta}\in\mathbb{R}^p} \min_{\boldsymbol{\theta}\in\mathbb{R}^n} \Big\{ \frac{1}{2n}\|\boldsymbol{Y} - \mathbf{X}^\top\boldsymbol{\beta} - \sqrt{n}\,\boldsymbol{\theta}\|_2^2 + \lambda_s\|\boldsymbol{\beta}\|_1 + \lambda_o\|\boldsymbol{\theta}\|_1 \Big\}, \tag{3}$$

where $\lambda_s, \lambda_o > 0$ are tuning parameters. This estimator is very attractive from a computational perspective, since it can be seen as the Lasso for the augmented design matrix $\mathbf{M} = [\mathbf{X}, \sqrt{n}\,\mathbf{I}_n]$, where $\mathbf{I}_n$ is the $n \times n$ identity matrix. To date, the best known rate for this type of estimator is

$$\sigma\Big(\frac{s\log p}{n}\Big)^{1/2} + \sigma\Big(\frac{o}{n}\Big)^{1/2}, \tag{4}$$

obtained in (Nguyen and Tran, 2013) under some restrictions on $(n, p, s, o)$. A quick comparison of (2) and (4) shows that the latter is sub-optimal. Indeed, the ratio of the two rates may be as large as $(n/o)^{1/2}$. The main goal of the present paper is to show that this sub-optimality is not an intrinsic property of the estimator (3), but rather an artefact of previous proof techniques. By using a refined argument, we prove that $\widehat{\boldsymbol{\beta}}$ defined by (3) does attain the optimal rate under very mild assumptions.

In the sequel, we refer to $\widehat{\boldsymbol{\beta}}$ as $\ell_1$-penalized Huber's $M$-estimator. The rationale for this term is that the minimization with respect to $\boldsymbol{\theta}$ in (3) can be done explicitly. It yields (Donoho and Montanari, 2016, Section 6)

$$\widehat{\boldsymbol{\beta}} \in \arg\min_{\boldsymbol{\beta}\in\mathbb{R}^p} \Big\{ \lambda_o^2 \sum_{i=1}^n \Phi\Big(\frac{y_i - \boldsymbol{X}_i^\top\boldsymbol{\beta}}{\lambda_o\sqrt{n}}\Big) + \lambda_s\|\boldsymbol{\beta}\|_1 \Big\}, \tag{5}$$

where $\Phi : \mathbb{R} \to \mathbb{R}$ is Huber's function defined by $\Phi(u) = (1/2)u^2 \wedge (|u| - 1/2)$.

To prove the rate-optimality of the estimator $\widehat{\boldsymbol{\beta}}$, we first establish a risk bound for a general design matrix $\mathbf{X}$ not necessarily formed by Gaussian vectors. This is done in the next section. Then, in Section 3, we state and discuss the result showing that all the necessary conditions are satisfied for the Gaussian design. Relevant prior work is presented in Section 4, while Section 5 discusses potential extensions. Section 7 provides a summary of our results and an outlook on future work. The proofs are deferred to the supplementary material.

## 2 Risk bound for the $\ell_1$-penalized Huber's $M$-estimator

This section is devoted to bringing forward sufficient conditions on the design matrix that allow for rate-optimal risk bounds for the estimator $\widehat{\beta}$ defined by (3) or, equivalently, by (5). There are two qualitative conditions that can be easily seen to be necessary: we call them restricted invertibility and incoherence. Indeed, even when there is no contamination, *i.e.*, the number of outliers is known to be $o = 0$, the matrix $\mathbf{X}$ has to satisfy a restricted invertibility condition (such as restricted isometry, restricted eigenvalue or compatibility) in order that the Lasso estimator (3) does achieve the optimal rate $\sigma\sqrt{(s/n)\log(p/s)}$. On the other hand, in the case where $n = p$ and $\mathbf{X} = \sqrt{n}\,\mathbf{I}_n$, even in the extremely favorable situation where the noise $\boldsymbol{\xi}$ is zero, the only identifiable vector is $\boldsymbol{\beta}^* + \boldsymbol{\theta}^*$. Therefore, it is impossible to consistently estimate $\boldsymbol{\beta}^*$ when the design matrix $\mathbf{X}$ is aligned with the identity matrix $\mathbf{I}_n$ or close to be so.

The next definition formalizes what we call restricted invertibility and incoherence by introducing three notions: the transfer principle, the incoherence property and the augmented transfer principle. We will show that these notions play a key role in robust estimation by $\ell_1$-penalized least squares.

**Definition 1.** *Let $\mathbf{Z} \in \mathbb{R}^{n \times p}$ be a (random) matrix and $\boldsymbol{\Sigma} \in \mathbb{R}^{p \times p}$. We use notation $\mathbf{Z}^{(n)} = \mathbf{Z}/\sqrt{n}$.*

    (i) *We say that $\mathbf{Z}$ satisfies the transfer principle with $\mathsf{a}_1 \in (0,1)$ and $\mathsf{a}_2 \in (0,\infty)$, denoted by $\mathrm{TP}_{\boldsymbol{\Sigma}}(\mathsf{a}_1; \mathsf{a}_2)$, if for all $\boldsymbol{v} \in \mathbb{R}^p$,*

$$\left\|\mathbf{Z}^{(n)}\boldsymbol{v}\right\|_2 \geq \mathsf{a}_1\|\boldsymbol{\Sigma}^{1/2}\boldsymbol{v}\|_2 - \mathsf{a}_2\|\boldsymbol{v}\|_1. \tag{6}$$

    (ii) *We say that $\mathbf{Z}$ satisfies the incoherence property $\mathrm{IP}_{\boldsymbol{\Sigma}}(\mathsf{b}_1; \mathsf{b}_2; \mathsf{b}_3)$ for some positive numbers $\mathsf{b}_1$, $\mathsf{b}_2$ and $\mathsf{b}_3$, if for all $[\boldsymbol{v}; \boldsymbol{u}] \in \mathbb{R}^{p+n}$,*

$$|\boldsymbol{u}^\top\mathbf{Z}^{(n)}\boldsymbol{v}| \leq \mathsf{b}_1\|\boldsymbol{\Sigma}^{1/2}\boldsymbol{v}\|_2\|\boldsymbol{u}\|_2 + \mathsf{b}_2\|\boldsymbol{v}\|_1\|\boldsymbol{u}\|_2 + \mathsf{b}_3\|\boldsymbol{\Sigma}^{1/2}\boldsymbol{v}\|_2\|\boldsymbol{u}\|_1.$$

    (iii) *We say that $\mathbf{Z}$ satisfies the augmented transfer principle $\mathrm{ATP}_{\boldsymbol{\Sigma}}(\mathsf{c}_1; \mathsf{c}_2; \mathsf{c}_3)$ for some positive numbers $\mathsf{c}_1$, $\mathsf{c}_2$ and $\mathsf{c}_3$, if for all $[\boldsymbol{v}; \boldsymbol{u}] \in \mathbb{R}^{p+n}$,*

$$\|\mathbf{Z}^{(n)}\boldsymbol{v} + \boldsymbol{u}\|_2 \geq \mathsf{c}_1\left\|[\boldsymbol{\Sigma}^{1/2}\boldsymbol{v}; \boldsymbol{u}]\right\|_2 - \mathsf{c}_2\|\boldsymbol{v}\|_1 - \mathsf{c}_3\|\boldsymbol{u}\|_1. \tag{7}$$

Note that the transfer principle was already well-known to be important in sparse estimation; a more general formulation of it can be found in (Juditsky and Nemirovski, 2011, Eq. 37). Note also that these three properties are inter-related and related to extreme singular values of the matrix $\mathbf{Z}^{(n)}$.

**(P1)** If $\mathbf{Z}$ satisfies $\mathrm{ATP}_{\boldsymbol{\Sigma}}(\mathsf{c}_1; \mathsf{c}_2; \mathsf{c}_3)$ then it also satisfies $\mathrm{TP}_{\boldsymbol{\Sigma}}(\mathsf{c}_1; \mathsf{c}_2)$.

**(P2)** If $\mathbf{Z}$ satisfies $\mathrm{TP}_{\boldsymbol{\Sigma}}(\mathsf{a}_1; \mathsf{a}_2)$ and $\mathrm{IP}_{\boldsymbol{\Sigma}}(\mathsf{b}_1; \mathsf{b}_2; \mathsf{b}_3)$ then it also satisfies $\mathrm{ATP}_{\boldsymbol{\Sigma}}(\mathsf{c}_1; \mathsf{c}_2; \mathsf{c}_3)$ with $\mathsf{c}_1^2 = \mathsf{a}_1^2 - \mathsf{b}_1 - \alpha^2$, $\mathsf{c}_2 = \mathsf{a}_2 + 2\mathsf{b}_2/\alpha$ and $\mathsf{c}_3 = 2\mathsf{b}_3/\alpha$ for any positive $\alpha < \sqrt{\mathsf{a}_1^2 - \mathsf{b}_1}$.

**(P3)** If $\mathbf{Z}$ satisfies $\mathrm{IP}_{\boldsymbol{\Sigma}}(\mathsf{b}_1; \mathsf{b}_2; \mathsf{b}_3)$, then it also satisfies $\mathrm{IP}_{\boldsymbol{\Sigma}}(0; \mathsf{b}_2; \mathsf{b}_1 + \mathsf{b}_3)$

**(P4)** Any matrix $\mathbf{Z}$ satisfies $\mathrm{TP}_{\mathbf{I}}(s_p(\mathbf{Z}^{(n)}); 0)$, and $\mathrm{IP}_{\mathbf{I}}(s_1(\mathbf{Z}^{(n)}); 0; 0)$, where $s_p(\mathbf{Z}^{(n)})$ and $s_1(\mathbf{Z}^{(n)})$ are, respectively, the $p$-th largest and the largest singular values of $\mathbf{Z}^{(n)}$.

Claim (P1) is true, since if we choose $\boldsymbol{u} = \boldsymbol{0}$ in (7) we obtain (6). Claim (P2) coincides with Lemma 7, proved in the supplement. (P3) is a direct consequence of the inequality $\|\boldsymbol{u}\|_2 \leq \|\boldsymbol{u}\|_1$, valid for any vector $\boldsymbol{u}$. (P4) is a well-known characterization of the smallest and the largest singular values of a matrix. We will show later on that a Gaussian matrix satisfies with high probability all these conditions with constants $\mathsf{a}_1$ and $\mathsf{c}_1$ independent of $(n, p)$ and $\mathsf{a}_2$, $\mathsf{b}_2$, $\mathsf{b}_3$, $\mathsf{c}_2$, $\mathsf{c}_3$ of order $n^{-1/2}$, up to logarithmic factors.

To state the main theorem of this section, we consider the simplified setting in which $\lambda_s = \lambda_o = \lambda$. Remind that in practice it is always recommended to normalize the columns of the matrix $\boldsymbol{X}$ so that their Euclidean norm is of the order $\sqrt{n}$. The more precise version of the next result with better constants is provided in the supplement (see Proposition 1). We recall that a matrix $\boldsymbol{\Sigma}$ is said to satisfy the restricted eigenvalue condition $\mathrm{RE}(s, c_0)$ with some constant $\varkappa > 0$, if $\|\boldsymbol{\Sigma}^{1/2}\boldsymbol{v}\|_2 \geq \varkappa\|\boldsymbol{v}_J\|_2$ for any vector $\boldsymbol{v} \in \mathbb{R}^p$ and any set $J \subset \{1, \ldots, p\}$ such that $\mathrm{Card}(J) \leq s$ and $\|\boldsymbol{v}_{J^c}\|_1 \leq c_0\|\boldsymbol{v}_J\|_1$.

**Theorem 1.** *Let $\boldsymbol{\Sigma}$ satisfy the $\mathrm{RE}(s, 5)$ condition with constant $\varkappa > 0$. Let $\mathsf{b}_1$, $\mathsf{b}_2$, $\mathsf{b}_3$, $\mathsf{c}_1$, $\mathsf{c}_2$, $\mathsf{c}_3$ be some positive real numbers such that $\mathbf{X}$ satisfies the $\mathrm{IP}_{\boldsymbol{\Sigma}}(0; \mathsf{b}_2; \mathsf{b}_3)$ and the $\mathrm{ATP}_{\boldsymbol{\Sigma}}(\mathsf{c}_1; \mathsf{c}_2; \mathsf{c}_3)$.*

*Assume that for some $\delta \in (0,1)$, the tuning parameter $\lambda$ satisfies*

$$\lambda\sqrt{n} \geq \sqrt{8\log(n/\delta)} \bigvee \big(\max_{j=1,\ldots,p} \|\mathbf{X}^{(n)}_{\bullet,j}\|_2\big)\sqrt{8\log(p/\delta)}.$$

*If the sparsity $s$ and the number of outliers $o$ satisfy the condition*

$$\frac{s}{\varkappa^2} + o \leq \frac{\mathsf{c}_1^2}{400\big(\mathsf{c}_2 \vee \mathsf{c}_3 \vee 5\mathsf{b}_2/\mathsf{c}_1\big)^2}, \tag{8}$$

*then, with probability at least $1 - 2\delta$, we have*

$$\big\|\mathbf{\Sigma}^{1/2}(\widehat{\boldsymbol{\beta}} - \boldsymbol{\beta}^*)\big\|_2 \leq \frac{24\lambda}{\mathsf{c}_1^2}\Big(\frac{2\mathsf{c}_2}{\mathsf{c}_1} \bigvee \frac{\mathsf{b}_3}{\mathsf{c}_1^2}\Big)\Big(\frac{s}{\varkappa^2} + 7o\Big) + \frac{5\lambda\sqrt{s}}{6\mathsf{c}_1^2\varkappa}. \tag{9}$$

Theorem 1 is somewhat hard to parse. At this stage, let us simply mention that in the case of a Gaussian design considered in the next section, $\mathsf{c}_1$ is of order 1 while $\mathsf{b}_2, \mathsf{b}_3, \mathsf{c}_2, \mathsf{c}_3$ are of order $n^{-1/2}$, up to a factor logarithmic in $p$, $n$ and $1/\delta$. Here $\delta$ is an upper bound on the probability that the Gaussian matrix $\mathbf{X}$ does not satisfy either $\mathrm{IP}_{\mathbf{\Sigma}}$ or $\mathrm{ATP}_{\mathbf{\Sigma}}$. Since Theorem 1 allows us to choose $\lambda$ of the order $\sqrt{\log\{(p+n)/\delta\}/n}$, we infer from (9) that the error of estimating $\boldsymbol{\beta}^*$, measured in Euclidean norm, is of order $\frac{s}{n\varkappa^2} + \frac{o}{n} + (\frac{s}{n\varkappa^2})^{1/2} = O(\frac{o}{n} + (\frac{s}{n\varkappa^2})^{1/2})$, under the assumption that $(\frac{s}{n\varkappa^2} + \frac{o}{n})\log(np/\delta)$ is smaller than a universal constant.

To complete this section, we present a sketch of the proof of Theorem 1. In order to convey the main ideas without diving too much into technical details, we assume $\mathbf{\Sigma} = \mathbf{I}_p$. This means that the RE condition is satisfied with $\varkappa = 1$ for any $s$ and $c_0$. From the fact that the $\mathrm{ATP}_{\mathbf{\Sigma}}$ holds for $\mathbf{X}$, we infer that $[\mathbf{X}\,\sqrt{n}\,\mathbf{I}_n]$ satisfies the $\mathrm{RE}(s+o, 5)$ condition with the constant $\mathsf{c}_1/2$. Using the well-known risk bounds for the Lasso estimator (Bickel et al., 2009), we get

$$\|\widehat{\boldsymbol{\beta}} - \boldsymbol{\beta}^*\|_2^2 + \|\widehat{\boldsymbol{\theta}} - \boldsymbol{\theta}^*\|_2^2 \leq C\lambda^2(s+o) \quad \text{and} \quad \|\widehat{\boldsymbol{\beta}} - \boldsymbol{\beta}^*\|_1 + \|\widehat{\boldsymbol{\theta}} - \boldsymbol{\theta}^*\|_1 \leq C\lambda(s+o). \tag{10}$$

Note that these are the risk bounds established in[3] (Candès and Randall, 2008; Dalalyan and Chen, 2012; Nguyen and Tran, 2013). These bounds are most likely unimprovable as long as the estimation of $\boldsymbol{\theta}^*$ is of interest. However, if we focus only on the estimation error of $\boldsymbol{\beta}^*$, considering $\boldsymbol{\theta}^*$ as a nuisance parameter, the following argument leads to a sharper risk bound. First, we note that

$$\widehat{\boldsymbol{\beta}} \in \arg\min_{\boldsymbol{\beta}\in\mathbb{R}^p} \Big\{\frac{1}{2n}\|\boldsymbol{Y} - \mathbf{X}\boldsymbol{\beta} - \sqrt{n}\,\widehat{\boldsymbol{\theta}}\|_2^2 + \lambda\|\boldsymbol{\beta}\|_1\Big\}.$$

The KKT conditions of this convex optimization problem take the following form

$$^1\!/_n\mathbf{X}^\top(\boldsymbol{Y} - \mathbf{X}\widehat{\boldsymbol{\beta}} - \sqrt{n}\,\widehat{\boldsymbol{\theta}}) \in \lambda\cdot\mathrm{sgn}(\widehat{\boldsymbol{\beta}}),$$

where $\mathrm{sgn}(\widehat{\boldsymbol{\beta}})$ is the subset of $\mathbb{R}^p$ containing all the vectors $\boldsymbol{w}$ such that $w_j\widehat{\beta}_j = |\widehat{\beta}_j|$ and $|w_j| \leq 1$ for every $j \in \{1,\ldots,p\}$. Multiplying the last displayed equation from left by $\boldsymbol{\beta}^* - \widehat{\boldsymbol{\beta}}$, we get

$$^1\!/_n(\boldsymbol{\beta}^* - \widehat{\boldsymbol{\beta}})^\top\mathbf{X}^\top(\boldsymbol{Y} - \mathbf{X}\widehat{\boldsymbol{\beta}} - \sqrt{n}\,\widehat{\boldsymbol{\theta}}) \leq \lambda\big(\|\boldsymbol{\beta}^*\|_1 - \|\widehat{\boldsymbol{\beta}}\|_1\big).$$

Recall now that $\boldsymbol{Y} = \mathbf{X}\boldsymbol{\beta}^* + \sqrt{n}\,\boldsymbol{\theta}^* + \boldsymbol{\xi}$ and set $\boldsymbol{v} = \boldsymbol{\beta}^* - \widehat{\boldsymbol{\beta}}$ and $\boldsymbol{u} = \boldsymbol{\theta}^* - \widehat{\boldsymbol{\theta}}$. We arrive at

$$^1\!/_n\|\mathbf{X}\boldsymbol{v}\|_2^2 = {}^1\!/_n\boldsymbol{v}^\top\mathbf{X}^\top\mathbf{X}\boldsymbol{v} \leq -\boldsymbol{v}^\top(\mathbf{X}^{(n)})^\top\boldsymbol{u} - {}^1\!/_n\boldsymbol{v}^\top\mathbf{X}^\top\boldsymbol{\xi} + \lambda\big(\|\boldsymbol{\beta}^*\|_1 - \|\widehat{\boldsymbol{\beta}}\|_1\big).$$

On the one hand, the duality inequality and the lower bound on $\lambda$ imply that $|\boldsymbol{v}^\top\mathbf{X}^\top\boldsymbol{\xi}| \leq \|\boldsymbol{v}\|_1\|\mathbf{X}^\top\boldsymbol{\xi}\|_\infty \leq n\lambda\|\boldsymbol{v}\|_1/2$. On the other hand, well-known arguments yield $\|\boldsymbol{\beta}^*\|_1 - \|\widehat{\boldsymbol{\beta}}\|_1 \leq 2\|\boldsymbol{v}_S\|_1 - \|\boldsymbol{v}\|_1$. Therefore, we have

$$^1\!/_n\|\mathbf{X}\boldsymbol{v}\|_2^2 \leq |\boldsymbol{v}^\top(\mathbf{X}^{(n)})^\top\boldsymbol{u}| + {}^\lambda\!/_2\big(4\|\boldsymbol{v}_S\|_1 - \|\boldsymbol{v}\|_1\big). \tag{11}$$

Since $\mathbf{X}$ satisfies the $\mathrm{ATP}_{\mathbf{I}}(\mathsf{c}_1, \mathsf{c}_2, \mathsf{c}_3)$ that implies the $\mathrm{TP}_{\mathbf{I}}(\mathsf{c}_1, \mathsf{c}_2)$, we get $\mathsf{c}_1^2\|\boldsymbol{v}\|_2^2 \leq {}^2\!/_n\|\mathbf{X}\boldsymbol{v}\|_2^2 + 2\mathsf{c}_2^2\|\boldsymbol{v}\|_1^2$. Combining with (11), this yields

$$\begin{aligned}
\mathsf{c}_1^2\|\boldsymbol{v}\|_2^2 \quad &\leq \quad 2|\boldsymbol{v}^\top(\mathbf{X}^{(n)})^\top\boldsymbol{u}| + \lambda\big(4\|\boldsymbol{v}_S\|_1 - \|\boldsymbol{v}\|_1\big) + 2\mathsf{c}_2^2\|\boldsymbol{v}\|_1^2 \\
&\overset{\mathrm{IP}_{\mathbf{I}}(0,\mathsf{b}_2,\mathsf{b}_3)}{\leq} \quad 2\mathsf{b}_3\|\boldsymbol{v}\|_2\|\boldsymbol{u}\|_1 + 2\mathsf{b}_2\|\boldsymbol{v}\|_1\|\boldsymbol{u}\|_2 + \lambda\big(4\|\boldsymbol{v}_S\|_1 - \|\boldsymbol{v}\|_1\big) + 2\mathsf{c}_2^2\|\boldsymbol{v}\|_1^2 \\
&\leq \quad \frac{\mathsf{c}_1^2}{2}\|\boldsymbol{v}\|_2^2 + \frac{2\mathsf{b}_3^2}{\mathsf{c}_1^2}\|\boldsymbol{u}\|_1^2 + \|\boldsymbol{v}\|_1(2\mathsf{b}_2\|\boldsymbol{u}\|_2 - \lambda) + 4\lambda\|\boldsymbol{v}_S\|_1 + 2\mathsf{c}_2^2\|\boldsymbol{v}\|_1^2.
\end{aligned} \tag{12}$$

Using the first inequality in (10) and condition (8), we upper bound $(2\mathsf{b}_2\|\boldsymbol{u}\|_2 - \lambda)$ by 0. To upper bound the second last term, we use the Cauchy-Schwarz inequality: $4\lambda\|\boldsymbol{v}_S\|_1 \le 4\lambda\sqrt{s}\,\|\boldsymbol{v}\|_2 \le (4/\mathsf{c}_1)^2\lambda^2 s + (\mathsf{c}_1/2)^2\|\boldsymbol{v}\|_2^2$. Combining all these bounds and rearranging the terms, we arrive at

$$(\mathsf{c}_1^2/4)\|\boldsymbol{v}\|_2^2 \le 2\{(\mathsf{b}_3/\mathsf{c}_1) \vee \mathsf{c}_2\}^2(\|\boldsymbol{u}\|_1 + \|\boldsymbol{v}\|_1)^2 + (4/\mathsf{c}_1)^2\lambda^2 s.$$

Taking the square root of both sides and using the second inequality in (10), we obtain an inequality of the same type as (9) but with slightly larger constants. As a concluding remark for this sketch of proof, let us note that if instead of using the last arguments, we replace all the error terms appearing in (12) by their upper bounds provided by (10), we do not get the optimal rate.

## 3 The case of Gaussian design

Our main result, Theorem 1, shows that if the design matrix satisfies the transfer principle and the incoherence property with suitable constants, then the $\ell_1$-penalized Huber's $M$-estimator achieves the optimal rate under adversarial contamination. As a concrete example of a design matrix for which the aforementioned conditions are satisfied, we consider the case of correlated Gaussian design. As opposed to most of prior work on robust estimation for linear regression with Gaussian design, we allow the covariance matrix to have a non degenerate null space. We will simply assume that the $n$ rows of the matrix $\mathbf{X}$ are independently drawn from the Gaussian distribution $\mathcal{N}_p(\mathbf{0}, \boldsymbol{\Sigma})$ with a covariance matrix $\boldsymbol{\Sigma}$ satisfying the $\mathrm{RE}(s,5)$ condition. We will also assume in this section that all the diagonal entries of $\boldsymbol{\Sigma}$ are equal to 1: $\boldsymbol{\Sigma}_{jj} = 1$. The more formal statements of the results, provided in the supplementary material, do not require this condition.

**Theorem 2.** *Let $\delta \in (0, 1/7)$ be a tolerance level and $n \ge 100$. For every positive semi-definite matrix $\boldsymbol{\Sigma}$ with all the diagonal entries bounded by one, with probability at least $1 - 2\delta$, the matrix $\mathbf{X}$ satisfies the $\mathrm{TP}_{\boldsymbol{\Sigma}}(\mathsf{a}_1, \mathsf{a}_2)$, the $\mathrm{IP}_{\boldsymbol{\Sigma}}(\mathsf{b}_1, \mathsf{b}_2, \mathsf{b}_3)$ and the $\mathrm{ATP}_{\boldsymbol{\Sigma}}(\mathsf{c}_1, \mathsf{c}_2, \mathsf{c}_3)$ with constants*

$$\mathsf{a}_1 = 1 - \frac{4.3 + \sqrt{2\log(9/\delta)}}{\sqrt{n}}, \qquad \mathsf{a}_2 = \mathsf{b}_2 = 1.2\sqrt{\frac{2\log p}{n}}$$

$$\mathsf{b}_1 = \frac{4.8\sqrt{2} + \sqrt{2\log(81/\delta)}}{\sqrt{n}}, \qquad \mathsf{b}_3 = 1.2\sqrt{\frac{2\log n}{n}},$$

$$\mathsf{c}_1 = \frac{3}{4} - \frac{17.5 + 9.6\sqrt{2\log(2/\delta)}}{\sqrt{n}}, \qquad \mathsf{c}_2 = 3.6\sqrt{\frac{2\log p}{n}}, \qquad \mathsf{c}_3 = 2.4\sqrt{\frac{2\log n}{n}}.$$

The proof of this result is provided in the supplementary material. It relies on by now standard tools such as Gordon's comparison inequality, Gaussian concentration inequality and the peeling argument. Note that the $\mathrm{TP}_{\boldsymbol{\Sigma}}$ and related results have been obtained in Raskutti et al. (2010); Oliveira (2016); Rudelson and Zhou (2013). The $\mathrm{IP}_{\boldsymbol{\Sigma}}$ is basically a combination of a high probability version of Chevet's inequality (Vershynin, 2018, Exercises 8.7.3-4) and the peeling argument. A property similar to the $\mathrm{ATP}_{\boldsymbol{\Sigma}}$ for Gaussian matrices with non degenerate covariance was established in (Nguyen and Tran, 2013, Lemma 1) under further restrictions on $n, p, s, o$.

**Theorem 3.** *There exist universal positive constants $\mathsf{d}_1$, $\mathsf{d}_2$, $\mathsf{d}_3$ such that if*

$$\frac{s\log p}{\varkappa^2} + o\log n \le \mathsf{d}_1 n \qquad and \qquad 1/7 \ge \delta \ge 2e^{-\mathsf{d}_2 n}$$

*then, with probability at least $1 - 4\delta$, $\ell_1$-penalized Huber's $M$-estimator with $\lambda_s^2 n = 9\sigma^2 \log(p/\delta)$ and $\lambda_o^2 n = 8\sigma^2 \log(n/\delta)$ satisfies*

$$\left\|\boldsymbol{\Sigma}^{1/2}(\widehat{\boldsymbol{\beta}} - \boldsymbol{\beta}^*)\right\|_2 \le \mathsf{d}_3\sigma\left\{\left(\frac{s\log(p/\delta)}{n\varkappa^2}\right)^{1/2} + \frac{o\log(n/\delta)}{n}\right\}. \tag{13}$$

Even though the constants appearing in Theorem 2 are reasonably small and smaller than in the analogous results in prior work, the constants $\mathsf{d}_1$, $\mathsf{d}_2$ and $\mathsf{d}_3$ are large, too large for being of any practical relevance. Finally, let us note that if $s$ and $o$ are known, it is very likely that following the techniques developed in (Bellec et al., 2018, Theorem 4.2), one can replace the terms $\log(p/\delta)$ and $\log(n/\delta)$ in (13) by $\log(p/s\delta)$ and $\log(n/o\delta)$, respectively.

Comparing Theorem 3 with (Nguyen and Tran, 2013, Theorem 1), we see that our rate improvement is not only in terms of its dependence on the proportion of outliers, $o/n$, but also in terms of the condition number $\varkappa$, which is now completely decoupled from $o$ in the risk bound.

While our main focus is on the high dimensional situation in which $p$ can be larger than $n$, it also applies to the case of small dimensional dense vectors, *i.e.*, when $s = p$ is significantly smaller than $n$. One of the applications of such a setting is the problem of stylized communication considered, for instance, in (Candès and Randall, 2008). The problem is to transmit a signal $\boldsymbol{\beta}^* \in \mathbb{R}^p$ to a remote receiver. What the receiver gets is a linearly transformed codeword $\mathbf{X}\boldsymbol{\beta}^*$ corrupted by small noise and malicious errors. While all the entries of the received codeword are affected by noise, only a fraction of them is corrupted by malicious errors, corresponding to outliers. The receiver has access to the corrupted version of $\mathbf{X}\boldsymbol{\beta}^*$ as well as to the encoding matrix $\mathbf{X}$. Theorem 3.1 from (Candès and Randall, 2008) establishes that the Dantzig selector (Candès and Tao, 2007), for a properly chosen tuning parameter proportional to the noise level, achieves the (sub-optimal) rate $\sigma^2(s + o)/n$, up to a logarithmic factor. A similar result, with a noise-level-free version of the Dantzig selector, was proved in (Dalalyan and Chen, 2012). Our Theorem 3 implies that the error of the $\ell_1$-penalized Huber's estimator goes to zero at the faster rate $\sigma^2\{(s/n) + (o/n)^2\}$. Finally, one can deduce from Theorem 3 that as soon as the number of outliers satisfies $o = o(\sqrt{sn/\varkappa^2})$, the rate of convergence remains the same as in the outlier-free setting.

# 4 Prior work

As attested by early references such as (Tukey, 1960), robust estimation has a long history. A remarkable—by now classic—result by Huber (1964) shows that among all the shift invariant $M$-estimators of a location parameter, the one that minimizes the asymptotic variance corresponds to the loss function $\phi(x) = 1/2\{x^2 \wedge (2x - 1)\}$. This result was proved in the case when the reference distribution is univariate Gaussian. Apart from some exceptions, such as (Yatracos, 1985), during several decades the literature on robust estimation was mainly exploring the notions of breakdown point, influence function, asymptotic efficiency, etc., see for instance (Donoho and Gasko, 1992; Hampel et al., 2005; Huber and Ronchetti, 2009) and the recent survey (Yu and Yao, 2017). A more recent trend in statistics is to focus on finite sample risk bounds that are minimax-rate-optimal when the sample size $n$, the dimension $p$ of the unknown parameter and the number $o$ of outliers tend jointly to infinity (Chen et al., 2018, 2016; Gao, 2017).

In the problem of estimating the mean of a multivariate Gaussian distribution, it was shown that the optimal rate of the estimation error measured in Euclidean norm scales as $(p/n)^{1/2} + (o/n)$. Similar results were established for the problem of robust linear regression as well. However, the estimator that was shown to achieve this rate under fairly general conditions on the design is based on minimizing regression depths, which is a hard computational problem. Several alternative robust estimators with polynomial complexity were proposed (Diakonikolas et al., 2016; Lai et al., 2016; Cheng et al., 2019; Collier and Dalalyan, 2017; Diakonikolas et al., 2018).

Many recent papers studied robust linear regression. (Karmalkar and Price, 2018) considered $\ell_1$-constrained minimization of the $\ell_1$-norm of residuals and found a sharp threshold on the proportion of outliers determining whether the error of estimation tends to zero or not, when the noise level goes to zero. From a methodological point of view, $\ell_1$-penalized Huber's estimator has been considered in (Sardy et al., 2001; She and Owen, 2011; Lee et al., 2012). These papers contain also comprehensive empirical evaluation and proposals for data-driven choice of tuning parameters. Robust sparse regression with an emphasis on contaminated design was investigated in (Chen et al., 2013; Balakrishnan et al., 2017; Diakonikolas et al., 2019; Liu et al., 2018, 2019). Iterative and adaptive hard thresholding approaches were considered in (Bhatia et al., 2015, 2017; Suggala et al., 2019). Methods based on penalizing the vector of outliers were studied by Li (2013); Foygel and Mackey (2014); Adcock et al. (2018), who adopted a more signal-processing point of view in which the noise vector is known to have a small $\ell_2$ norm and nothing else is known about it. We should stress that our proof techniques share many common features with those in (Foygel and Mackey, 2014).

The problem of robust estimation of graphical models, closely related to the present work, was addressed in (Balmand and Dalalyan, 2015; Katiyar et al., 2019; Liu et al., 2019). Quite surprisingly,

at least to us, the minimax rate of robust estimation of the precision matrix in Frobenius norm is not known yet.

# 5 Extensions

The results presented in previous sections pave the way for some future investigations, that are discussed below. None of these extensions is carried out in this work, they are listed here as possible avenues for future research.

**Contaminated design**  In addition to labels, the features also might be corrupted by outliers. This is the case, for instance, in Gaussian graphical models. Formally, this means that instead of observing the clean data $\{(\boldsymbol{X}_i^\circ, y_i^\circ); i = 1, \ldots, n\}$ satisfying $y_i^\circ = (\boldsymbol{X}_i^\circ)^\top \beta^* + \xi_i$, we observe $\{(\boldsymbol{X}_i, y_i); i = 1, \ldots, n\}$ such that $(\boldsymbol{X}_i, y_i) = (\boldsymbol{X}_i^\circ, y_i^\circ)$ for all $i$ except for a fraction of outliers $i \in O$. In such a setting, we can set $\theta_i^* = (y_i - \boldsymbol{X}_i^\top \boldsymbol{\beta}^* - \xi_i)/\sqrt{n}$ and recover exactly the same model as in (1).

The important difference as compared to the setting investigated in previous section is that it is not reasonable anymore to assume that the feature vectors $\{\boldsymbol{X}_i : i \in O\}$ are iid Gaussian. In the adversarial setting, they may even be correlated with the noise vector $\boldsymbol{\xi}$. It is then natural to remove all the observations for which $\max_j |\boldsymbol{X}_{ij}| > \sqrt{2 \log np/\delta}$ and to assume, that the $\ell_1$-penalized Huber estimator is applied to data for which $\max_{ij} |\boldsymbol{X}_{ij}| \leq \sqrt{2 \log np/\delta}$. This implies that $\lambda$ can be chosen of the order of[4] $\sigma \tilde{O}(n^{-1/2} + (o/n))$, which is an upper bound on $\|\boldsymbol{X}^\top \boldsymbol{\xi}\|_\infty / n$.

In addition, $\mathrm{TP}_{\boldsymbol{\Sigma}}$ is clearly satisfied since it is satisfied for the submatrix $\mathbf{X}_{O^c}$ and $\|\mathbf{X}\boldsymbol{v}\|_2 \geq \|\mathbf{X}_{O^c}\boldsymbol{v}\|_2$. As for the $\mathrm{IP}_{\boldsymbol{\Sigma}}$, we know from Theorem 2 that $\mathbf{X}_{O^c}$ satisfies $\mathrm{IP}_{\boldsymbol{\Sigma}}$ with constants $\mathsf{b}_1$, $\mathsf{b}_2$, $\mathsf{b}_3$ of order $\tilde{O}(n^{-1/2})$. On the other hand,

$$|\boldsymbol{u}_O^\top \mathbf{X}_O \boldsymbol{v}| \leq \|\mathbf{X}\|_\infty \|\boldsymbol{u}_O\|_1 \|\boldsymbol{v}\|_1 \leq \sqrt{2o \log(np/\delta)} \|\boldsymbol{u}_O\|_2 \|\boldsymbol{v}\|_1.$$

This implies that $\mathbf{X}$ satisfies $\mathrm{IP}_{\boldsymbol{\Sigma}}$ with $\mathsf{b}_1 = \tilde{O}(n^{-1/2})$, $\mathsf{b}_2 = \tilde{O}((o/n)^{1/2})$ and $\mathsf{b}_3 = \tilde{O}(n^{-1/2})$. Applying Theorem 1, we obtain that if $(so + o^2) \log(np) \leq cn$ for a sufficiently small constant $c > 0$, then with high probability

$$\|\boldsymbol{\Sigma}^{1/2}(\widehat{\boldsymbol{\beta}} - \boldsymbol{\beta}^*)\|_2 = \sigma \tilde{O}\left\{ \sqrt{\frac{s}{n}} + \frac{o\sqrt{s}}{n} + \sqrt{\frac{o}{n}}\left(\frac{1}{\sqrt{n}} + \frac{o}{n}\right)(s + o) \right\} = \sigma O\left\{ \sqrt{\frac{s}{n}} + \frac{\sqrt{o^3}}{n} \right\}.$$

This rate of convergence appear to be slower than those obtained by methods tailored to deal with corruption in design, see (Liu et al., 2018, 2019) and the references therein. Using more careful analysis, this rate might be improvable. On the positive side, unlike many of its competitors, the estimator $\widehat{\boldsymbol{\beta}}$ has the advantage of being independent of the covariance matrix $\boldsymbol{\Sigma}$ and on the sparsity $s$. Furthermore, the upper bound does not depend, even logarithmically, on $\|\boldsymbol{\beta}^*\|_2$. Finally, if $o^3 \leq sn$, our bound yields the minimax-optimal rate. To the best of our knowledge, none of the previously studied robust estimators has such a property.

**Sub-Gaussian design**  The proof of Theorem 2 makes use of some results, such as Gordon-Sudakov-Fernique or Gaussian concentration inequality, which are specific to the Gaussian distribution. A natural question is whether the rate $\sigma\{(\frac{s \log(p/s)}{n})^{1/2} + \frac{o}{n}\}$ can be obtained for more general design distributions. In the case of a sub-Gaussian design with the scale- parameter 1, it should be possible to adapt the methodology developed in this work to show that the $\mathrm{TP}_{\boldsymbol{\Sigma}}$ and the $\mathrm{IP}_{\boldsymbol{\Sigma}}$ are satisfied with high-probability. Indeed, for proving the $\mathrm{IP}_{\boldsymbol{\Sigma}}$, it is possible to replace Gordon's comparison inequality by Talagrand's sub-Gaussian comparison inequality (Vershynin, 2018, Cor. 8.6.2). The Gaussian concentration inequality can be replaced by generic chaining.

**Heavier tailed noise distributions**  For simplicity, we assumed in the paper that the random variables $\xi_i$ are drawn from a Gaussian distribution. As usual for the Lasso analysis, all the results extend to the case of sub-Gaussian noise, see (Koltchinskii, 2011). Indeed, we only need to control tail probabilities of the random variable $\|\mathbf{X}^\top \boldsymbol{\xi}\|_\infty$ and $\|\boldsymbol{\xi}\|_\infty$, which can be done using standard tools.

We believe that it is possible to extend our results beyond sub-Gaussian noise, by assuming some type of heavy-tailed distributions. The rationale behind this is that any random variable $\xi$ can be written (in many different ways) as a sum of a sub-Gaussian variable $\xi^{\text{noise}}$ and a "sparse" variable $\xi^{\text{out}}$. By "sparse" we mean that $\xi^{\text{out}}$ takes the value 0 with high probability. The most naive way for getting such a decomposition is to set $\xi^{\text{noise}} = \xi\mathbb{1}(|\xi| < \tau)$ and $\xi^{\text{out}} = \xi\mathbb{1}(|\xi| \geq \tau)$. The random noise terms $\xi_i^{\text{out}}$ can be merged with $\theta_i$ and considered as outliers. We hope that this approach can establish a connection between two types of robustness: robustness to outliers considered in this work and robustness to heavy tails considered in many recent papers (Devroye et al., 2016; Catoni, 2012; Minsker, 2018; Lugosi and Mendelson, 2019; Lecué and Lerasle, 2017).

## 6 Numerical illustration

We performed a synthetic experiment to illustrate the obtained theoretical result and to check that it is in line with numerical results. We chose $n = 1000$ and $p = 100$ for 3 different levels of sparsity $s = 5, 15, 25$. The noise variance was set to 1 and $\boldsymbol{\beta}^*$ was set to have its first $s$ non-zero coordinates equal to 10. Each corrupted response coordinate was $\theta_j^* = 10$. The fraction $\epsilon = o/n$ of outliers was ranging between 0 and 0.25 with a step-size of 5 for the number of outliers $o$ is used. The MSE was computed using 200 independent repetitions. The optimisation problem in (3) was solved using the `glmnet` package with the tuning parameters $\lambda_s = \lambda_o = \sqrt{(8/n)(\log(p/s) + \log(n/o))}$.

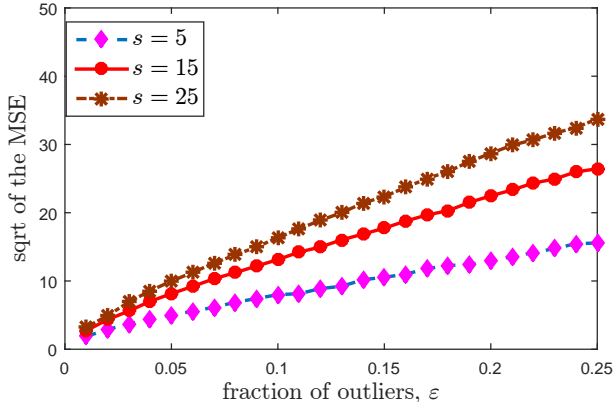

The obtained plots clearly demonstrate that there is a linear dependence on $\varepsilon$ of the square-root of the mean squared error.

## 7 Conclusion

We provided the first proof of the rate-optimality—up to logarithmic terms that can be avoided—of $\ell_1$-penalized Huber's $M$-estimator in the setting of robust linear regression with adversarial contamination. We established this result under the assumption that the design is Gaussian with a covariance matrix $\boldsymbol{\Sigma}$ that need not be invertible. The condition number governing the risk bound is the ratio of the largest diagonal entry of $\Sigma$ and its restricted eigenvalue. Thus, in addition to improving the rate of convergence, we also relaxed the assumptions on the design. Furthermore, we outlined some possible extensions, namely to corrupted design and/or sub-Gaussian design, which seem to be fairly easy to carry out building on the current work.

Next on our agenda is the more thorough analysis of the robust estimation by $\ell_1$-penalization in the case of contaminated design. A possible approach, complementary to the one described in Section 5 above, is to adopt an errors-in-variables point of view similar to that developed in (Belloni et al., 2016). Another interesting avenue for future research is the development of scale-invariant robust estimators and their adaptation to the Gaussian graphical models. This can be done using methodology brought forward in (Sun and Zhang, 2013; Balmand and Dalalyan, 2015). Finally, we would like to better understand what is the largest fraction of outliers for which the $\ell_1$-penalized Huber's $M$-estimator has a risk—measured in Euclidean norm—upper bounded by $\sigma o/n$. Answering this question even under stringent assumptions of independent standard Gaussian design $\boldsymbol{X}_{ij}$ with $(s \log p)/n$ going to zero as $n$ tends to infinity would be of interest.

# 8 Acknowledgements

We would like to thank the reviewers for the careful reading of the paper and for helpful and thoughtful remarks. This work was supported by the grants Investissements d'Avenir ANR-11IDEX-0003/Labex Ecodec/ANR11-LABX-0047 and ANR-11-LABX-0056-LMH, Labex LMH.

## Footnotes

[1]In the sequel, we use notation $\|\boldsymbol{\beta}\|_q = (\sum_j |\beta_j|^q)^{1/q}$ for any vector $\boldsymbol{\beta} \in \mathbb{R}^p$ and any $q \geq 1$.

[2]In the sense that there is no algorithm computing these estimators in time polynomial in $(n, p, s, o)$.

[3]the first two references deal with the small dimensional case only, that is where $s = p \ll n$.

[4]We use notation $a_n = \tilde{O}(b_n)$ as a shorthand for $a_n \leq C b_n \log^c n$ for some $C, c > 0$ and for every $n$.

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
