[Supplementary Material]

# Supplementary material

The theorems stated in the paper are consequences of Proposition 2, Proposition 4 and Proposition 3. These results are proved in subsequent sections, which are organized as follows. Section 8 contains tight risk bounds for general matrices satisfying the transfer principle and the incoherence property. We then show in Section 9 that the Gaussian design satisfies, with high probability, both the transfer principle and the incoherence property. We complete the paper by showing how Theorem 1, Theorem 2 and Theorem 3 can be deduced from Proposition 2, Proposition 4 and Proposition 3.

To help the reader to navigate through the proof without losing the thread, the diagram below outlines the relations between different auxiliary results.

Thus, Proposition 1 establishes a risk bound valid under $\text{ATP}_{\boldsymbol{\Sigma}}$. This risk bound is sub-optimal for Gaussian designs, but it is an intermediate step for getting the final risk bound, established in Proposition 2. The latter follows from the $\text{TP}_{\boldsymbol{\Sigma}}$, $\text{IP}_{\boldsymbol{\Sigma}}$ and an auxiliary result proved in Lemma 3. The fact that the $\text{TP}_{\boldsymbol{\Sigma}}$ holds true for Gaussian matrices is proved in Proposition 3 as a consequence of Lemma 3 and one-parameter peeling (Lemma 5). Similarly, the fact that the $\text{IP}_{\boldsymbol{\Sigma}}$ holds true for Gaussian matrices is proved in Proposition 4 as a consequence of Lemma 4 and two-parameter peeling (Lemma 6).

# 8 Main technical results for general design matrices

In the sequel, we denote by $\mathbb{S}^{k-1}$ the unit sphere in $\mathbb{R}^k$ with respect to the Euclidean norm centered at the origin. With a slight abuse of notation, $\mathbb{R}^k$ will be identified with $\mathbb{R}^{k \times 1}$. The unit ball with respect to the $\ell_p$-norm centered at the origin will be denoted by $\mathbb{B}_p^k$. Given a matrix $\boldsymbol{\Sigma} \in \mathbb{R}^{p \times p}$, we will use the definition $\rho(\boldsymbol{\Sigma}) := \max_{j \in [p]} \sqrt{\boldsymbol{\Sigma}_{jj}}$ without further notice. We will use notation $\boldsymbol{\Delta}^{\boldsymbol{\beta}} = \widehat{\boldsymbol{\beta}} - \boldsymbol{\beta}^*$, $\boldsymbol{\Delta}^{\boldsymbol{\theta}} = \widehat{\boldsymbol{\theta}} - \boldsymbol{\theta}^*$ and $\boldsymbol{\Delta} = [\boldsymbol{\Delta}^{\boldsymbol{\beta}}; \boldsymbol{\Delta}^{\boldsymbol{\theta}}] \in \mathbb{R}^{p+n}$. We denote by $S$ the support of $\boldsymbol{\beta}^*$ and by $O$ that of $\boldsymbol{\theta}^*$. We know that $\text{Card}(S) \le s$ and $\text{Card}(O) \le o$. Throughout, we set $\gamma = \lambda_s / \lambda_o$ and define the dimension reduction cone $\mathcal{C}_{S,O}(c_0, \gamma) = \{(\boldsymbol{u}, \boldsymbol{v}) \in \mathbb{R}^n \times \mathbb{R}^p : \|\boldsymbol{u}_{O^c}\|_1 + \gamma \|\boldsymbol{v}_{S^c}\|_1 \le c_0(\|\boldsymbol{u}_O\|_1 + \gamma \|\boldsymbol{v}_S\|_1)\}$, where $c_0 \ge 1$ is a constant.

## 8.1 Augmented transfer principle implies the sub-optimal rate

This section is devoted to the proof of the fact that the estimators $\widehat{\boldsymbol{\beta}}$ and $\widehat{\boldsymbol{\theta}}$ achieve, up to logarithmic factors, the rates

$$\frac{s}{n \varkappa^2} + \frac{o}{n} \qquad \text{and} \qquad \frac{s}{\sqrt{n} \varkappa^2} + \frac{o}{\sqrt{n}}$$

for squared $\ell_2$ error and $\ell_1$ errors, respectively. This is true under suitable conditions on the design matrix $\mathbf{X}$. These rates are not optimal, but they will help us to obtain the optimal rates.

**Proposition 1.** *Let $\boldsymbol{\Sigma}$ satisfy the $\mathrm{RE}(s,5)$ with constant $\varkappa > 0$. Let $\mathsf{c}_1, \mathsf{c}_2, \mathsf{c}_3$ and $\gamma$ be some positive real numbers satisfying*

$$8\big(\mathsf{c}_2 \vee \gamma\mathsf{c}_3\big)\left(\frac{s}{\varkappa^2} + \frac{6.25o}{\gamma^2}\right)^{1/2} \leq \mathsf{c}_1.$$

*Assume that on some event $\Omega$, the following conditions are met:*

(i) $\mathbf{X}$ *satisfies the* $\mathrm{ATP}_{\boldsymbol{\Sigma}}\left(\mathsf{c}_1; \mathsf{c}_2; \mathsf{c}_3\right)$.

(ii) $\lambda_s = \gamma\lambda_o \geq (2/n)\|\mathbf{X}^\top\boldsymbol{\xi}\|_\infty, \quad and \quad \lambda_o \geq (2/\sqrt{n})\|\boldsymbol{\xi}\|_\infty.$

*Then, on the same event $\Omega$, we have $\boldsymbol{\Delta} \in \mathcal{C}_{S,O}(3, \lambda_s/\lambda_o)$ and*

$$\big\|\boldsymbol{\Sigma}^{1/2}\boldsymbol{\Delta}^{\boldsymbol{\beta}}\big\|_2^2 + \big\|\boldsymbol{\Delta}^{\boldsymbol{\theta}}\big\|_2^2 \leq \frac{36}{\mathsf{c}_1^4}\left(\frac{\lambda_s^2 s}{\varkappa^2} + 6.25\lambda_o^2 o\right),$$

$$\lambda_s\big\|\boldsymbol{\Delta}^{\boldsymbol{\beta}}\big\|_1 + \lambda_o\big\|\boldsymbol{\Delta}^{\boldsymbol{\theta}}\big\|_1 \leq \frac{24}{\mathsf{c}_1^2}\left(\frac{\lambda_s^2 s}{\varkappa^2} + 6.25\lambda_o^2 o\right). \tag{14}$$

*Proof.* First, we use the KKT conditions to infer that for some vectors $\boldsymbol{u} \in \mathbb{B}_\infty^n$ and $\boldsymbol{v} \in \mathbb{B}_\infty^p$ such that $\boldsymbol{u}^\top\widehat{\boldsymbol{\theta}} = \|\widehat{\boldsymbol{\theta}}\|_1$ and $\boldsymbol{v}^\top\widehat{\boldsymbol{\beta}} = \|\widehat{\boldsymbol{\beta}}\|_1$, we have

$$[\mathbf{X}^{(n)}\,\mathbf{I}_n]^\top\big(\boldsymbol{y}^{(n)} - \mathbf{X}^{(n)}\widehat{\boldsymbol{\beta}} - \widehat{\boldsymbol{\theta}}\big) = [\lambda_s\boldsymbol{v}; \lambda_o\boldsymbol{u}].$$

Using the facts that $\boldsymbol{y}^{(n)} = \mathbf{X}^{(n)}\boldsymbol{\beta}^* + \boldsymbol{\theta}^* + \boldsymbol{\xi}^{(n)}$ and rearranging the terms, the last display takes the form

$$[\mathbf{X}^{(n)}\,\mathbf{I}_n]^\top[\mathbf{X}^{(n)}\,\mathbf{I}_n]\boldsymbol{\Delta} = [(\mathbf{X}^{(n)})^\top\boldsymbol{\xi}^{(n)}\,; \boldsymbol{\xi}^{(n)}] + [\lambda_s\boldsymbol{v}; \lambda_o\boldsymbol{u}].$$

Multiplying the last display from the left by $\boldsymbol{\Delta}^\top$, we arrive at

$$\|[\mathbf{X}^{(n)}\,\mathbf{I}_n]\boldsymbol{\Delta}\|_2^2 = (\boldsymbol{\Delta}^{\boldsymbol{\beta}})^\top(\mathbf{X}^{(n)})^\top\boldsymbol{\xi}^{(n)} + (\boldsymbol{\Delta}^{\boldsymbol{\theta}})^\top\boldsymbol{\xi}^{(n)} + \lambda_s(\boldsymbol{\Delta}^{\boldsymbol{\beta}})^\top\boldsymbol{v} + \lambda_o(\boldsymbol{\Delta}^{\boldsymbol{\theta}})^\top\boldsymbol{u}.$$

The relations $\|\boldsymbol{v}\|_\infty \leq 1$ and $\boldsymbol{v}^\top\widehat{\boldsymbol{\beta}} = \|\widehat{\boldsymbol{\beta}}\|_1$ imply that $(\boldsymbol{\Delta}^{\boldsymbol{\beta}})^\top\boldsymbol{v} = (\boldsymbol{\beta}^* - \widehat{\boldsymbol{\beta}})^\top\boldsymbol{v} = (\boldsymbol{\beta}^*)^\top\boldsymbol{v} - \|\widehat{\boldsymbol{\beta}}\|_1 \leq \|\boldsymbol{\beta}^*\|_1 - \|\widehat{\boldsymbol{\beta}}\|_1$. Similarly, $(\boldsymbol{\Delta}^{\boldsymbol{\theta}})^\top\boldsymbol{u} \leq \|\boldsymbol{\theta}^*\|_1 - \|\widehat{\boldsymbol{\theta}}\|_1$. Combining these bounds with the duality inequality and the last display, we infer that

$$\|[\mathbf{X}^{(n)}\,\mathbf{I}_n]\boldsymbol{\Delta}\|_2^2 \leq \|\boldsymbol{\Delta}^{\boldsymbol{\beta}}\|_1\|(\mathbf{X}^{(n)})^\top\boldsymbol{\xi}^{(n)}\|_\infty + \|\boldsymbol{\Delta}^{\boldsymbol{\theta}}\|_1\|\boldsymbol{\xi}^{(n)}\|_\infty$$
$$+ \lambda_s\big(\|\boldsymbol{\beta}^*\|_1 - \|\widehat{\boldsymbol{\beta}}\|_1\big) + \lambda_o\big(\|\boldsymbol{\theta}^*\|_1 - \|\widehat{\boldsymbol{\theta}}\|_1\big)$$
$$\overset{\text{(ii)}}{\leq} (\lambda_s/2)\|\boldsymbol{\Delta}^{\boldsymbol{\beta}}\|_1 + (\lambda_o/2)\|\boldsymbol{\Delta}^{\boldsymbol{\theta}}\|_1 + \lambda_s\big(\|\boldsymbol{\beta}^*\|_1 - \|\widehat{\boldsymbol{\beta}}\|_1\big) + \lambda_o\big(\|\boldsymbol{\theta}^*\|_1 - \|\widehat{\boldsymbol{\theta}}\|_1\big). \tag{15}$$

Recall that $J = \{j : \boldsymbol{\beta}_j \neq 0\}$ and $O = \{i : \boldsymbol{\theta}_i^* \neq 0\}$. We have

$$\|\boldsymbol{\Delta}^{\boldsymbol{\beta}}\|_1 + 2\|\boldsymbol{\beta}^*\|_1 - 2\|\widehat{\boldsymbol{\beta}}\|_1 = \|\boldsymbol{\Delta}^{\boldsymbol{\beta}}\|_1 + 2\|\boldsymbol{\beta}_S^*\|_1 - 2\|\widehat{\boldsymbol{\beta}}_S\|_1 - 2\|\boldsymbol{\Delta}_{S^c}^{\boldsymbol{\beta}}\|_1$$
$$\leq \|\boldsymbol{\Delta}^{\boldsymbol{\beta}}\|_1 + 2\|\boldsymbol{\Delta}_S^{\boldsymbol{\beta}}\|_1 - 2\|\boldsymbol{\Delta}_{S^c}^{\boldsymbol{\beta}}\|_1$$
$$= 3\|\boldsymbol{\Delta}_S^{\boldsymbol{\beta}}\|_1 - \|\boldsymbol{\Delta}_{S^c}^{\boldsymbol{\beta}}\|_1.$$

The same type of reasoning leads to $\|\boldsymbol{\Delta}^{\boldsymbol{\theta}}\|_1 + 2\|\boldsymbol{\theta}^*\|_1 - 2\|\widehat{\boldsymbol{\theta}}\|_1 \leq 3\|\boldsymbol{\Delta}_O^{\boldsymbol{\theta}}\|_1 - \|\boldsymbol{\Delta}_{O^c}^{\boldsymbol{\theta}}\|_1$. Combining these inequalities with (15), we get

$$\|[\mathbf{X}^{(n)}\,\mathbf{I}_n]\boldsymbol{\Delta}\|_2^2 \leq (\lambda_s/2)\big(3\|\boldsymbol{\Delta}_S^{\boldsymbol{\beta}}\|_1 - \|\boldsymbol{\Delta}_{S^c}^{\boldsymbol{\beta}}\|_1\big) + (\lambda_o/2)\big(3\|\boldsymbol{\Delta}_O^{\boldsymbol{\theta}}\|_1 - \|\boldsymbol{\Delta}_{O^c}^{\boldsymbol{\theta}}\|_1\big).$$

On the one hand, since the left hand side is non negative, this obviously implies that the vector $\boldsymbol{\Delta}$ belongs to the dimension reduction cone $\mathcal{C}_{S,O}(3, \gamma)$. On the other hand, using the $\mathrm{ATP}_{\boldsymbol{\Sigma}}$,

$$\mathsf{c}_1\big\|[\boldsymbol{\Sigma}^{1/2}\boldsymbol{\Delta}^{\boldsymbol{\beta}}\,; \boldsymbol{\Delta}^{\boldsymbol{\theta}}]\big\|_2 - \mathsf{c}_2\|\boldsymbol{\Delta}^{\boldsymbol{\beta}}\|_1 - \mathsf{c}_3\|\boldsymbol{\Delta}^{\boldsymbol{\theta}}\|_1$$
$$\leq \sqrt{(\lambda_s/2)\big(3\|\boldsymbol{\Delta}_S^{\boldsymbol{\beta}}\|_1 - \|\boldsymbol{\Delta}_{S^c}^{\boldsymbol{\beta}}\|_1\big) + (\lambda_o/2)\big(3\|\boldsymbol{\Delta}_O^{\boldsymbol{\theta}}\|_1 - \|\boldsymbol{\Delta}_{O^c}^{\boldsymbol{\theta}}\|_1\big)}. \tag{16}$$

We split the rest of the proof into two parts: the first corresponds to the case $5\|\boldsymbol{\Delta}_S^{\boldsymbol{\beta}}\|_1 \geq \|\boldsymbol{\Delta}_{S^c}^{\boldsymbol{\beta}}\|_1$ while the second treats the case $5\|\boldsymbol{\Delta}_S^{\boldsymbol{\beta}}\|_1 \leq \|\boldsymbol{\Delta}_{S^c}^{\boldsymbol{\beta}}\|_1$. The main goal of this splitting is to avoid imposing strong assumption on $\boldsymbol{\Sigma}$ such as $\sigma_{\min}(\boldsymbol{\Sigma}) > 0$ and to use the RE condition only.

**Case 1:** $5\|\mathbf{\Delta}_S^{\boldsymbol{\beta}}\|_1 \geq \|\mathbf{\Delta}_{S^c}^{\boldsymbol{\beta}}\|_1$. This is the simple case, since we know that $\mathbf{\Delta}^{\boldsymbol{\beta}}$ lies in the suitable dimension reduction cone for which we can use the RE condition. We first use the already proved fact $\mathbf{\Delta} \in \mathcal{C}_{S,O}(3, \gamma)$ to infer that

$$
\begin{aligned}
\mathsf{c}_2\|\mathbf{\Delta}^{\boldsymbol{\beta}}\|_1 + \mathsf{c}_3\|\mathbf{\Delta}^{\boldsymbol{\theta}}\|_1 &\leq \left(\frac{\mathsf{c}_2}{\lambda_s} \bigvee \frac{\mathsf{c}_3}{\lambda_o}\right)(\lambda_s\|\mathbf{\Delta}^{\boldsymbol{\beta}}\|_1 + \lambda_o\|\mathbf{\Delta}^{\boldsymbol{\theta}}\|_1) \\
&\leq 4\left(\frac{\mathsf{c}_2}{\lambda_s} \bigvee \frac{\mathsf{c}_3}{\lambda_o}\right)(\lambda_s\|\mathbf{\Delta}_S^{\boldsymbol{\beta}}\|_1 + \lambda_o\|\mathbf{\Delta}_O^{\boldsymbol{\theta}}\|_1) \\
&\leq 4\left(\frac{\mathsf{c}_2}{\lambda_s} \bigvee \frac{\mathsf{c}_3}{\lambda_o}\right)\left(\frac{\lambda_s^2 s}{\varkappa^2} + \lambda_o^2 o\right)^{1/2}(\varkappa^2\|\mathbf{\Delta}_S^{\boldsymbol{\beta}}\|_2^2 + \|\mathbf{\Delta}_O^{\boldsymbol{\theta}}\|_2^2)^{1/2} \\
&\leq 4\left(\frac{\mathsf{c}_2}{\lambda_s} \bigvee \frac{\mathsf{c}_3}{\lambda_o}\right)\left(\frac{\lambda_s^2 s}{\varkappa^2} + \lambda_o^2 o\right)^{1/2}\left\|[\mathbf{\Sigma}^{1/2}\mathbf{\Delta}^{\boldsymbol{\beta}} ; \mathbf{\Delta}^{\boldsymbol{\theta}}]\right\|_2. \quad (17)
\end{aligned}
$$

Similarly, the right hand side of (16) can be bounded by the square-root of the expression

$$
\begin{aligned}
3(\lambda_s/2)\|\mathbf{\Delta}_S^{\boldsymbol{\beta}}\|_1 + 3(\lambda_o/2)\|\mathbf{\Delta}_O^{\boldsymbol{\theta}}\|_1 &\leq 1.5\left(\frac{\lambda_s^2 s}{\varkappa^2} + \lambda_o^2 o\right)^{1/2}(\varkappa^2\|\mathbf{\Delta}_S^{\boldsymbol{\beta}}\|_2^2 + \|\mathbf{\Delta}_O^{\boldsymbol{\theta}}\|_2^2)^{1/2} \\
&\leq 1.5\left(\frac{\lambda_s^2 s}{\varkappa^2} + \lambda_o^2 o\right)^{1/2}\left\|[\mathbf{\Sigma}^{1/2}\mathbf{\Delta}^{\boldsymbol{\beta}} ; \mathbf{\Delta}^{\boldsymbol{\theta}}]\right\|_2. \quad (18)
\end{aligned}
$$

To ease notation, we define $A = 4\left(\frac{\mathsf{c}_2}{\lambda_s} \bigvee \frac{\mathsf{c}_3}{\lambda_o}\right)\left(\frac{\lambda_s^2 s}{\varkappa^2} + \lambda_o^2 o\right)^{1/2}$, $B = 1.5\left(\frac{\lambda_s^2 s}{\varkappa^2} + \lambda_o^2 o\right)^{1/2}$ and $x = \left\|[\mathbf{\Sigma}^{1/2}\mathbf{\Delta}^{\boldsymbol{\beta}} ; \mathbf{\Delta}^{\boldsymbol{\theta}}]\right\|_2$. These notations are valid in this proof only. From (16), (17), (18), we get

$$
\mathsf{c}_1 x \leq A x + \sqrt{B x} \implies x \leq \frac{B}{(\mathsf{c}_1 - A)^2}
$$

provided that $A \leq \mathsf{c}_1$. Assuming $2A \leq \mathsf{c}_1$, we get

$$
\left\|\mathbf{\Sigma}^{1/2}\mathbf{\Delta}^{\boldsymbol{\beta}}\right\|_2^2 + \left\|\mathbf{\Delta}^{\boldsymbol{\theta}}\right\|_2^2 \leq \frac{16B^2}{\mathsf{c}_1^4}.
$$

For deriving the bound on the $\ell_1$ norms of the errors, we first use the fact that $\mathbf{\Delta}$ lies in the dimension reduction cone, followed by the Cauchy-Schwarz inequality, to get

$$
\begin{aligned}
\lambda_s\left\|\mathbf{\Delta}^{\boldsymbol{\beta}}\right\|_1 + \lambda_o\|\mathbf{\Delta}^{\boldsymbol{\theta}}\|_1 &\leq 4(\lambda_s\|\mathbf{\Delta}_S^{\boldsymbol{\beta}}\|_1 + \lambda_o\|\mathbf{\Delta}_O^{\boldsymbol{\theta}}\|_1) \\
&\leq 4\left(\frac{\lambda_s^2 s}{\varkappa^2} + \lambda_o^2 o\right)^{1/2}\left\|[\mathbf{\Sigma}^{1/2}\mathbf{\Delta}^{\boldsymbol{\beta}} ; \mathbf{\Delta}^{\boldsymbol{\theta}}]\right\|_2 \\
&\leq \frac{16B}{\mathsf{c}_1^2}\left(\frac{\lambda_s^2 s}{\varkappa^2} + \lambda_o^2 o\right)^{1/2} \\
&= \frac{24}{\mathsf{c}_1^2}\left(\frac{\lambda_s^2 s}{\varkappa^2} + \lambda_o^2 o\right).
\end{aligned}
$$

**Case 2:** $5\|\mathbf{\Delta}_S^{\boldsymbol{\beta}}\|_1 < \|\mathbf{\Delta}_{S^c}^{\boldsymbol{\beta}}\|_1$. In this case, we can infer from the already proved fact $\mathbf{\Delta} \in \mathcal{C}_{S,O}(3, \gamma)$ that

$$
2\gamma\|\mathbf{\Delta}_S^{\boldsymbol{\beta}}\|_1 + \|\mathbf{\Delta}_{O^c}^{\boldsymbol{\theta}}\|_1 \leq 3\|\mathbf{\Delta}_O^{\boldsymbol{\theta}}\|_1.
$$

Hence, we have

$$
\begin{aligned}
\mathsf{c}_2\|\mathbf{\Delta}^{\boldsymbol{\beta}}\|_1 + \mathsf{c}_3\|\mathbf{\Delta}^{\boldsymbol{\theta}}\|_1 &\leq \left(\frac{\mathsf{c}_2}{\lambda_s} \bigvee \frac{\mathsf{c}_3}{\lambda_o}\right)(\lambda_s\|\mathbf{\Delta}^{\boldsymbol{\beta}}\|_1 + \lambda_o\|\mathbf{\Delta}^{\boldsymbol{\theta}}\|_1) \\
&\leq 4\left(\frac{\mathsf{c}_2}{\lambda_s} \bigvee \frac{\mathsf{c}_3}{\lambda_o}\right)(\lambda_s\|\mathbf{\Delta}_S^{\boldsymbol{\beta}}\|_1 + \lambda_o\|\mathbf{\Delta}_O^{\boldsymbol{\theta}}\|_1) \\
&\leq 10\left(\frac{\mathsf{c}_2}{\lambda_s} \bigvee \frac{\mathsf{c}_3}{\lambda_o}\right)\lambda_o\|\mathbf{\Delta}_O^{\boldsymbol{\theta}}\|_1 \\
&\leq 10\left(\frac{\mathsf{c}_2}{\lambda_s} \bigvee \frac{\mathsf{c}_3}{\lambda_o}\right)\lambda_o\sqrt{o}\,\|\mathbf{\Delta}^{\boldsymbol{\theta}}\|_2. \quad (19)
\end{aligned}
$$

Similarly, the right hand side of (16) can be bounded by the square-root of the expression

$$3(\lambda_s/2)\|\boldsymbol{\Delta}_S^{\boldsymbol{\beta}}\|_1 + 3(\lambda_o/2)\|\boldsymbol{\Delta}_O^{\boldsymbol{\theta}}\|_1 \leq (15/4)\lambda_o\|\boldsymbol{\Delta}_O^{\boldsymbol{\theta}}\|_1 \leq (15/4)\lambda_o\sqrt{o}\,\|\boldsymbol{\Delta}^{\boldsymbol{\theta}}\|_2. \qquad (20)$$

To ease notation, we define $A' = 10\left(\frac{c_2}{\lambda_s} \vee \frac{c_3}{\lambda_o}\right)\lambda_o\sqrt{o}$, $B' = (15/4)\lambda_o\sqrt{o}$ and $x' = \left\|[\boldsymbol{\Sigma}^{1/2}\boldsymbol{\Delta}^{\boldsymbol{\beta}}\,;\,\boldsymbol{\Delta}^{\boldsymbol{\theta}}]\right\|_2$. These notations are valid in this proof only. From (16), (19), (20), we get

$$\mathsf{c}_1 x' \leq A'x' + \sqrt{B'x'} \quad \implies \quad x' \leq \frac{B'}{(\mathsf{c}_1 - A')^2} \leq \frac{4B'}{\mathsf{c}_1^2}$$

provided that $2A' \leq \mathsf{c}_1$. Thus, we have proved the inequality

$$\left\|\boldsymbol{\Sigma}^{1/2}\boldsymbol{\Delta}^{\boldsymbol{\beta}}\right\|_2 \vee \left\|\boldsymbol{\Delta}^{\boldsymbol{\theta}}\right\|_2 \leq \frac{15\lambda_o\sqrt{o}}{\mathsf{c}_1^2},$$

which implies that

$$\gamma\left\|\boldsymbol{\Delta}^{\boldsymbol{\beta}}\right\|_1 + \left\|\boldsymbol{\Delta}^{\boldsymbol{\theta}}\right\|_1 \leq 4(\gamma\left\|\boldsymbol{\Delta}_J^{\boldsymbol{\beta}}\right\|_1 + \left\|\boldsymbol{\Delta}_O^{\boldsymbol{\theta}}\right\|_1) \leq 10\left\|\boldsymbol{\Delta}_O^{\boldsymbol{\theta}}\right\|_1 \leq 10\sqrt{o}\left\|\boldsymbol{\Delta}_O^{\boldsymbol{\theta}}\right\|_2 \leq \frac{150\lambda_o o}{\mathsf{c}_1^2}.$$

To complete the proof, it suffices to remark that the upper bounds provided in the statement of the proposition are larger than the bounds we have just established both in case 1 and in case 2. □

## 8.2 Augmented transfer principle and incoherence imply the nearly optimal rate

**Lemma 1.** *The following bound holds:*

$$\|\mathbf{X}^{(n)}\boldsymbol{\Delta}^{\boldsymbol{\beta}}\|_2^2 \leq (\boldsymbol{\Delta}^{\boldsymbol{\beta}})^\top(\mathbf{X}^{(n)})^\top\boldsymbol{\Delta}^{\boldsymbol{\theta}} + \|\boldsymbol{\Delta}^{\boldsymbol{\beta}}\|_1\|(\mathbf{X}^{(n)})^\top\boldsymbol{\xi}^{(n)}\|_\infty + \lambda_s\left(2\|\boldsymbol{\Delta}_S^{\boldsymbol{\beta}}\|_1 - \|\boldsymbol{\Delta}^{\boldsymbol{\beta}}\|_1\right).$$

*Proof.* We note that

$$\widehat{\boldsymbol{\beta}} \in \operatorname*{argmin}_{\boldsymbol{\beta}}\left\{\frac{1}{2}\left\|\boldsymbol{y}^{(n)} - \mathbf{X}^{(n)}\boldsymbol{\beta} - \widehat{\boldsymbol{\theta}}\right\|_2^2 + \lambda_s\|\boldsymbol{\beta}\|_1\right\}.$$

The KKT conditions of the above minimization problem imply that, for some $\boldsymbol{v} \in \mathbb{R}^p$ such that $\|\boldsymbol{v}\|_\infty \leq 1$ and $\boldsymbol{v}^\top\widehat{\boldsymbol{\beta}} = \|\widehat{\boldsymbol{\beta}}\|_1$,

$$\begin{aligned}\mathbf{0} &= (\mathbf{X}^{(n)})^\top\left(\mathbf{X}^{(n)}\widehat{\boldsymbol{\beta}} + \widehat{\boldsymbol{\theta}} - \boldsymbol{y}^{(n)}\right) + \lambda_s\boldsymbol{v} \\ &= (\mathbf{X}^{(n)})^\top\left(\mathbf{X}^{(n)}\boldsymbol{\Delta}^{\boldsymbol{\beta}} + \boldsymbol{\Delta}^{\boldsymbol{\theta}} - \boldsymbol{\xi}^{(n)}\right) + \lambda_s\boldsymbol{v}.\end{aligned}$$

Multiplying the above equality from the left by $(\boldsymbol{\Delta}^{\boldsymbol{\beta}})^\top$ we obtain

$$0 = \|\mathbf{X}^{(n)}\boldsymbol{\Delta}^{\boldsymbol{\beta}}\|_2^2 + (\boldsymbol{\Delta}^{\boldsymbol{\beta}})^\top(\mathbf{X}^{(n)})^\top\boldsymbol{\Delta}^{\boldsymbol{\theta}} - (\boldsymbol{\Delta}^{\boldsymbol{\beta}})^\top(\mathbf{X}^{(n)})^\top\boldsymbol{\xi}^{(n)} + \lambda_s(\widehat{\boldsymbol{\beta}} - \boldsymbol{\beta}^*)^\top\boldsymbol{v}.$$

From the above inequality, $\boldsymbol{v}^\top\widehat{\boldsymbol{\beta}} = \|\boldsymbol{\beta}\|_1$ and the fact that $\boldsymbol{v}^\top\boldsymbol{\beta}^* \leq \|\boldsymbol{\beta}^*\|_1$ (since $\|\boldsymbol{v}\|_\infty \leq 1$), we obtain that

$$\|\mathbf{X}^{(n)}\boldsymbol{\Delta}^{\boldsymbol{\beta}}\|_2^2 \leq -(\boldsymbol{\Delta}^{\boldsymbol{\beta}})^\top(\mathbf{X}^{(n)})^\top\boldsymbol{\Delta}^{\boldsymbol{\theta}} + \|\boldsymbol{\Delta}^{\boldsymbol{\beta}}\|_1\|(\mathbf{X}^{(n)})^\top\boldsymbol{\xi}^{(n)}\|_\infty + \lambda_s\left(\|\boldsymbol{\beta}^*\|_1 - \|\widehat{\boldsymbol{\beta}}\|_1\right).$$

One checks that

$$\|\boldsymbol{\beta}^*\|_1 - \|\widehat{\boldsymbol{\beta}}\|_1 \leq \|\boldsymbol{\Delta}_S^{\boldsymbol{\beta}}\|_1 - \|\boldsymbol{\Delta}_{S^c}^{\boldsymbol{\beta}}\|_1 = 2\|\boldsymbol{\Delta}_S^{\boldsymbol{\beta}}\|_1 - \|\boldsymbol{\Delta}^{\boldsymbol{\beta}}\|_1.$$

Combining this and the previous inequality we get the claim of the lemma. □

**Proposition 2.** *Let $\boldsymbol{\Sigma}$ satisfy the $\mathrm{RE}(s,5)$ with constant $\varkappa > 0$. Let $\mathsf{a}_1$, $\mathsf{a}_2$, $\mathsf{a}_3$, $\mathsf{b}_1$, $\mathsf{b}_2$, $\mathsf{c}_1$, $\mathsf{c}_2$, $\mathsf{c}_3$ and $\gamma$ be some positive real numbers satisfying*

$$8\left(\mathsf{c}_2 \vee \gamma\mathsf{c}_3\right)\left(\frac{s}{\varkappa^2} + \frac{6.25o}{\gamma^2}\right)^{1/2} \leq \mathsf{c}_1 \qquad (21)$$

$$36\mathsf{b}_2\left(\frac{s}{\varkappa^2} + \frac{6.25o}{\gamma^2}\right)^{1/2} \leq \mathsf{c}_1^2. \qquad (22)$$

*Assume that on some event $\Omega$, the following conditions are met:*

518       (i) $\mathbf{X}$ *satisfies the* $\mathrm{TP}_{\boldsymbol{\Sigma}}\,(\mathsf{a}_1; \mathsf{a}_2)$.

519      (ii) $\mathbf{X}$ *satisfies the* $\mathrm{IP}_{\boldsymbol{\Sigma}}\,(\mathsf{b}_1; \mathsf{b}_2; \mathsf{b}_3)$ .

520      (iii) $\mathbf{X}$ *satisfies the* $\mathrm{ATP}_{\boldsymbol{\Sigma}}\,(\mathsf{c}_1; \mathsf{c}_2; \mathsf{c}_3)$ .

521      (iv) $\lambda_s = \gamma\lambda_o \geq (2/n)\|\mathbf{X}^\top\boldsymbol{\xi}\|_\infty, \quad and \quad \lambda_o \geq (2/\sqrt{n})\|\boldsymbol{\xi}\|_\infty.$

522 *Then, on the same event $\Omega$, we have*

$$\left\|\boldsymbol{\Sigma}^{1/2}(\widehat{\boldsymbol{\beta}} - \boldsymbol{\beta}^*)\right\|_2 \leq \frac{24\lambda_s}{\mathsf{c}_1^2}\left(\frac{2\mathsf{a}_2}{\mathsf{a}_1} \bigvee \frac{(\mathsf{b}_1 + \mathsf{b}_3)\gamma}{\mathsf{a}_1^2}\right)\left(\frac{s}{\varkappa^2} + \frac{6.25o}{\gamma^2}\right) + \frac{5\lambda_s\sqrt{s}}{6\mathsf{a}_1^2\varkappa}.$$

523 *Proof.* Assume that we the event $\Omega$ is realized. Condition (21) implies that the claims of Proposition 1
524 hold true. In particular, the Euclidean norm of the error of estimating $\boldsymbol{\theta}^*$ can be bounded as follows:

$$\|\boldsymbol{\Delta}^{\boldsymbol{\theta}}\|_2 \leq \frac{6}{\mathsf{c}_1^2}\left(\frac{\lambda_s^2 s}{\varkappa^2} + 6.25\lambda_o^2 o\right)^{1/2} \leq \frac{\lambda_s}{6\mathsf{b}_2}, \tag{23}$$

525 where the last inequality follows from (22). Lemma 1 and item (ii) imply that

$$\|\mathbf{X}^{(n)}\boldsymbol{\Delta}^{\boldsymbol{\beta}}\|_2^2 \leq (\boldsymbol{\Delta}^{\boldsymbol{\beta}})^\top(\mathbf{X}^{(n)})^\top\boldsymbol{\Delta}^{\boldsymbol{\theta}} + \|\boldsymbol{\Delta}^{\boldsymbol{\beta}}\|_1\|(\mathbf{X}^{(n)})^\top\boldsymbol{\xi}^{(n)}\|_\infty + \lambda_s\left(2\|\boldsymbol{\Delta}_S^{\boldsymbol{\beta}}\|_1 - \|\boldsymbol{\Delta}^{\boldsymbol{\beta}}\|_1\right)$$

$$\overset{(iv)}{\leq} (\boldsymbol{\Delta}^{\boldsymbol{\beta}})^\top(\mathbf{X}^{(n)})^\top\boldsymbol{\Delta}^{\boldsymbol{\theta}} + \frac{\lambda_s}{2}\|\boldsymbol{\Delta}^{\boldsymbol{\beta}}\|_1 + \lambda_s\left(2\|\boldsymbol{\Delta}_S^{\boldsymbol{\beta}}\|_1 - \|\boldsymbol{\Delta}^{\boldsymbol{\beta}}\|_1\right)$$

$$\overset{\mathrm{IP}_{\boldsymbol{\Sigma}}}{\leq} \mathsf{b}_1\|\boldsymbol{\Sigma}^{1/2}\boldsymbol{\Delta}^{\boldsymbol{\beta}}\|_2\|\boldsymbol{\Delta}^{\boldsymbol{\theta}}\|_2 + \mathsf{b}_3\|\boldsymbol{\Sigma}^{1/2}\boldsymbol{\Delta}^{\boldsymbol{\beta}}\|_2\|\boldsymbol{\Delta}^{\boldsymbol{\theta}}\|_1 + 2\lambda_s\|\boldsymbol{\Delta}_S^{\boldsymbol{\beta}}\|_1 - \frac{\lambda_s}{3}\|\boldsymbol{\Delta}^{\boldsymbol{\beta}}\|_1$$

$$+ \mathsf{b}_2\|\boldsymbol{\Delta}^{\boldsymbol{\beta}}\|_1\|\boldsymbol{\Delta}^{\boldsymbol{\theta}}\|_2 - \frac{\lambda_s}{6}\|\boldsymbol{\Delta}^{\boldsymbol{\beta}}\|_1$$

$$\leq \mathsf{b}_1\|\boldsymbol{\Sigma}^{1/2}\boldsymbol{\Delta}^{\boldsymbol{\beta}}\|_2\|\boldsymbol{\Delta}^{\boldsymbol{\theta}}\|_2 + \mathsf{b}_3\|\boldsymbol{\Sigma}^{1/2}\boldsymbol{\Delta}^{\boldsymbol{\beta}}\|_2\|\boldsymbol{\Delta}^{\boldsymbol{\theta}}\|_1 + (\lambda_s/3)\left(5\|\boldsymbol{\Delta}_S^{\boldsymbol{\beta}}\|_1 - \|\boldsymbol{\Delta}_{S^c}^{\boldsymbol{\beta}}\|_1\right)$$

526 where the last line follows from the fact that $2\|\boldsymbol{\Delta}_S^{\boldsymbol{\beta}}\|_1 - 1/3\|\boldsymbol{\Delta}^{\boldsymbol{\beta}}\|_1 = 1/3(5\|\boldsymbol{\Delta}_S^{\boldsymbol{\beta}}\|_1 - \|\boldsymbol{\Delta}_{S^c}^{\boldsymbol{\beta}}\|_1)$
527 and (23). To ease notation, let us use notations $A = \mathsf{b}_1\|\boldsymbol{\Delta}^{\boldsymbol{\theta}}\|_2 + \mathsf{b}_3\|\boldsymbol{\Delta}^{\boldsymbol{\theta}}\|_1$, $B = \lambda_s/3(5\|\boldsymbol{\Delta}_S^{\boldsymbol{\beta}}\|_1 -$
528 $\|\boldsymbol{\Delta}_{S^c}^{\boldsymbol{\beta}}\|_1)_+$ and $x = \|\boldsymbol{\Sigma}^{1/2}\boldsymbol{\Delta}^{\boldsymbol{\beta}}\|_2$, which are valid for this proof only. On the one hand, combining
529 the last inequality and the $\mathrm{TP}_{\boldsymbol{\Sigma}}$, we arrive at

$$(\mathsf{a}_1 x - \mathsf{a}_2\|\boldsymbol{\Delta}^{\boldsymbol{\beta}}\|_1)_+^2 \leq Ax + B.$$

530 This implies that either $x \leq (\mathsf{a}_2/\mathsf{a}_1)\|\boldsymbol{\Delta}^{\boldsymbol{\beta}}\|_1$ or

$$\left(\mathsf{a}_1 x - \mathsf{a}_2\|\boldsymbol{\Delta}^{\boldsymbol{\beta}}\|_1 - \frac{A}{2\mathsf{a}_1}\right)^2 \leq B + \frac{A^2}{4\mathsf{a}_1^2} + \frac{A\mathsf{a}_2}{\mathsf{a}_1}\|\boldsymbol{\Delta}^{\boldsymbol{\beta}}\|_1.$$

531 Therefore, in both cases,

$$x \leq \frac{\mathsf{a}_2}{\mathsf{a}_1}\|\boldsymbol{\Delta}^{\boldsymbol{\beta}}\|_1 + \frac{A}{2\mathsf{a}_1^2} + \frac{1}{\mathsf{a}_1}\left\{B + \frac{A^2}{4\mathsf{a}_1^2} + \frac{A\mathsf{a}_2}{\mathsf{a}_1}\|\boldsymbol{\Delta}^{\boldsymbol{\beta}}\|_1\right\}^{1/2} \leq \frac{2\mathsf{a}_2}{\mathsf{a}_1}\|\boldsymbol{\Delta}^{\boldsymbol{\beta}}\|_1 + \frac{A}{\mathsf{a}_1^2} + \frac{B^{1/2}}{\mathsf{a}_1}. \tag{24}$$

532 On the other hand, the $\mathrm{RE}(s,5)$ property yields

$$B \leq \frac{5\lambda_s\|\boldsymbol{\Delta}_S^{\boldsymbol{\beta}}\|_1}{3} \leq \frac{5\lambda_s\sqrt{s}\|\boldsymbol{\Delta}_S^{\boldsymbol{\beta}}\|_2}{3} \leq \frac{5\lambda_s\sqrt{s}\,x}{3\varkappa} \leq \left(\frac{\mathsf{a}_1 x}{2} + \frac{5\lambda_s\sqrt{s}}{6\mathsf{a}_1\varkappa}\right)^2. \tag{25}$$

533 Combining (24) and (25), we get

$$\frac{x}{2} \leq \frac{2\mathsf{a}_2}{\mathsf{a}_1}\|\boldsymbol{\Delta}^{\boldsymbol{\beta}}\|_1 + \frac{A}{\mathsf{a}_1^2} + \frac{5\lambda_s\sqrt{s}}{6\mathsf{a}_1^2\varkappa}.$$

534 Replacing $A$ and $x$ by their expressions, we arrive at

$$\|\boldsymbol{\Sigma}^{1/2}\boldsymbol{\Delta}^{\boldsymbol{\beta}}\|_2 \leq \frac{2\mathsf{a}_2}{\mathsf{a}_1}\|\boldsymbol{\Delta}^{\boldsymbol{\beta}}\|_1 + \frac{\mathsf{b}_1\|\boldsymbol{\Delta}^{\boldsymbol{\theta}}\|_2 + \mathsf{b}_3\|\boldsymbol{\Delta}^{\boldsymbol{\theta}}\|_1}{\mathsf{a}_1^2} + \frac{5\lambda_s\sqrt{s}}{6\mathsf{a}_1^2\varkappa}$$

$$\leq \frac{2\mathsf{a}_2}{\mathsf{a}_1}\|\boldsymbol{\Delta}^{\boldsymbol{\beta}}\|_1 + \frac{\mathsf{b}_1 + \mathsf{b}_3}{\mathsf{a}_1^2}\|\boldsymbol{\Delta}^{\boldsymbol{\theta}}\|_1 + \frac{5\lambda_s\sqrt{s}}{6\mathsf{a}_1^2\varkappa}$$

$$\leq \left(\frac{2\mathsf{a}_2}{\gamma\mathsf{a}_1} \bigvee \frac{\mathsf{b}_1 + \mathsf{b}_3}{\mathsf{a}_1^2}\right)\left(\gamma\|\boldsymbol{\Delta}^{\boldsymbol{\beta}}\|_1 + \|\boldsymbol{\Delta}^{\boldsymbol{\theta}}\|_1\right) + \frac{5\lambda_s\sqrt{s}}{6\mathsf{a}_1^2\varkappa}.$$

Finally, combining inequality (14) from Proposition 1 with the last display we obtain

$$\left\|\mathbf{\Sigma}^{1/2}\mathbf{\Delta}^{\beta}\right\|_2 \leq \frac{24\lambda_o}{\mathsf{c}_1^2}\left(\frac{2\mathsf{a}_2}{\gamma\mathsf{a}_1}\bigvee\frac{\mathsf{b}_1+\mathsf{b}_3}{\mathsf{a}_1^2}\right)\left(\frac{\gamma^2 s}{\varkappa^2}+6.25o\right)+\frac{5\lambda_s\sqrt{s}}{6\mathsf{a}_1^2\varkappa}.$$

This completes the proof of the proposition. $\qquad\square$

# 9 Properties of Gaussian matrices

The next lemma ensures that the parameters $\lambda$ and $\gamma$ satisfy, with high-probability, condition ii) of Proposition 1 (which is the same as (iv) of Proposition 2).

**Lemma 2.** *Let the rows of $\mathbf{Z}$ be iid Gaussian with zero mean and covariance matrix $\mathbf{\Sigma}$ and $\boldsymbol{\xi}\sim\mathcal{N}_n(\mathbf{0},\sigma^2\mathbf{I}_n)$. Then the following two claims hold true.*

    *(i) For any $\delta\in(0,1]$, with probability at least $1-\delta$,*

$$\max_{j\in[p]}\|\mathbf{Z}_{\bullet,j}^{(n)}\|_2 \leq \left\{1+\sqrt{\frac{2\log(p/\delta)}{n}}\right\}\rho(\mathbf{\Sigma}).$$

    *(ii) For any $\delta\in(0,1]$ and $n\geq 2\log(3p/\delta)$, penalization factors such that*

$$\lambda_o\geq 2\sigma\sqrt{\frac{2\log(3n/\delta)}{n}},\qquad \lambda_s\geq 2\sigma\rho(\mathbf{\Sigma})\sqrt{\frac{2\log(3p/\delta)}{n}}\left(1+\sqrt{\frac{2\log(3p/\delta)}{n}}\right),$$

    *satisfy conditions of item (iv) of Proposition 2 with probability at least $1-\delta$.*

*Proof.* Let $\widetilde{\mathbf{Z}}:=\mathbf{Z}\mathbf{\Sigma}^{-1/2}$. We also note that

$$\|\mathbf{Z}_{\bullet,j}\|_2^2 = \sum_{i\in[n]}\left[\widetilde{\mathbf{Z}}_{i,\bullet}(\mathbf{\Sigma}^{1/2})_{\bullet,j}\right]^2,$$

where $\widetilde{\mathbf{Z}}_{1,\bullet}(\mathbf{\Sigma}^{1/2})_{\bullet,j},\dots,\widetilde{\mathbf{Z}}_{n,\bullet}(\mathbf{\Sigma}^{1/2})_{\bullet,j}$ are iid $\mathcal{N}(0,\mathbf{\Sigma}_{jj})$. By standard $\chi^2$ concentration inequalities, for all $j\in[p]$, with probability at least $1-\delta/p$,

$$\|\mathbf{Z}_{\bullet,j}^{(n)}\|_2 \leq \mathbf{\Sigma}_{jj}^{1/2}\left\{1+\sqrt{\frac{2\log(p/\delta)}{n}}\right\}.$$

Item (i) follows from this inequality using the union bound.

We now prove item (ii). Recall that $\mathbf{Z}$ and $\boldsymbol{\xi}\sim\mathcal{N}_n(0,\sigma^2\mathbf{I}_n)$ are independent and, therefore, conditionally on $\mathbf{Z}$, $(\mathbf{Z}_{\bullet,j})^{\top}\boldsymbol{\xi}\sim\mathcal{N}_n(0,\sigma^2\|\mathbf{Z}_{\bullet,j}\|_2^2)$. The well known maximal Gaussian concentration inequality implies that for all $j\in[p]$, with probability at least $1-\delta/3p$,

$$|(\mathbf{Z}_{\bullet,j}^{(n)})^{\top}\boldsymbol{\xi}^{(n)}| \leq \sigma\|\mathbf{Z}_{\bullet,j}^{(n)}\|\sqrt{\frac{2\log(3p/\delta)}{n}}. \tag{26}$$

Similarly, with probability at least $1-\delta/3$,

$$\|\boldsymbol{\xi}^{(n)}\|_{\infty} \leq \sigma\sqrt{\frac{2\log(3n/\delta)}{n}}. \tag{27}$$

Taking the union bound over the $p$ sets satisfying (26), the set satisfying (27) and the set satisfying item (i), we prove item (ii). $\qquad\square$

## 9.1 Bounding extrema on compact sets

In what follows, we will use the notion of Gaussian width for measuring the richness of a set of vectors. For a compact set $\mathcal{B}\subset\mathbb{R}^p$, we define the Gaussian width of $\mathcal{B}$ by

$$\mathscr{G}(\mathcal{B}) := \mathbb{E}\left[\sup_{\boldsymbol{b}\in\mathcal{B}}\boldsymbol{b}^{\top}\boldsymbol{\xi}\right],\qquad \boldsymbol{\xi}_i\overset{\text{iid}}{\sim}\mathcal{N}(0,1).$$

In view of (Boucheron et al., 2013, Theorem 2.5), for every symmetric $p \times p$ matrix $\mathbf{A}$, $\mathbb{E}\left[\|\mathbf{A}\boldsymbol{\xi}\|_\infty\right] \leq \{\max_{j \in [p]} (\mathbf{A}^2)_{jj}^{1/2}\}\sqrt{2\log p}$. This implies that

$$\mathscr{G}(\mathbf{A}\mathbb{B}_1^p) = \mathbb{E}[\|\mathbf{A}\boldsymbol{\xi}\|_\infty] \leq \rho(\mathbf{A}^2)\sqrt{2\log p}. \tag{28}$$

The above inequality is tight for orthogonal matrices $\mathbf{A}$, but it might be sub-optimal, up to a log factor, especially for poorly conditioned matrices $\mathbf{A}$.

**Lemma 3.** *Let $\mathbf{Z}$ be a $n \times p$ matrix with iid $\mathcal{N}(0,1)$ entries. For all $n \geq 1$, $t > 0$ and any compact set $\mathcal{B} \subset \mathbb{S}^{p-1}$, with probability at least $1 - \exp(-t^2/2)$,*

$$\inf_{\boldsymbol{b}\in\mathcal{B}} \|\mathbf{Z}\boldsymbol{b}\|_2 \geq \frac{n}{\sqrt{n+1}} - \mathscr{G}(\mathcal{B}) - t.$$

*As a consequence, for all $n \geq 1$ and $\delta \in (0,1]$, with with probability at least $1 - \delta$, the following inequality holds:*

$$\inf_{\boldsymbol{b}\in\mathcal{B}} \|\mathbf{Z}^{(n)}\boldsymbol{b}\|_2 \geq 1 - \frac{1}{2n} - \sqrt{\frac{2\log(1/\delta)}{n}} - \frac{\mathscr{G}(\mathcal{B})}{\sqrt{n}}.$$

*Proof.* The norm of $\mathbf{Z}\boldsymbol{b}$ can be written as

$$\|\mathbf{Z}\boldsymbol{b}\|_2 = \sup_{\boldsymbol{v}\in\mathbb{B}_2^n} \boldsymbol{v}^\top \mathbf{Z}\boldsymbol{b}.$$

We define the centered Gaussian process $Z_{\boldsymbol{b},\boldsymbol{v}} = -\boldsymbol{v}^\top\mathbf{Z}\boldsymbol{b} = -\sum_{i=1}^n \mathbf{Z}_i \boldsymbol{b} v_i$. It satisfies

$$\mathbb{E}[(Z_{\boldsymbol{b},\boldsymbol{v}} - Z_{\boldsymbol{b}',\boldsymbol{v}'})^2] = \|\boldsymbol{b}\boldsymbol{v}^\top - \boldsymbol{b}'(\boldsymbol{v}')^\top\|_F^2.$$

We are interested in upper bounding the quantity $\inf_{\boldsymbol{v}} \sup_{\boldsymbol{b}} Z_{\boldsymbol{b},\boldsymbol{v}}$. To this end, we define the process

$$W_{\boldsymbol{b},\boldsymbol{v}} = \text{trace}[\boldsymbol{v}^\top\boldsymbol{\xi}] + \text{trace}[\boldsymbol{b}^\top\bar{\boldsymbol{\xi}}],$$

where $\boldsymbol{\xi} \in \mathbb{R}^n$ and $\bar{\boldsymbol{\xi}} \in \mathbb{R}^p$ are two independent vectors with iid $\mathcal{N}(0,1)$ entries. One checks that

$$\mathbb{E}[(Z_{\boldsymbol{b},\boldsymbol{v}} - Z_{\boldsymbol{b}',\boldsymbol{v}'})^2] - \mathbb{E}[(W_{\boldsymbol{b},\boldsymbol{v}} - W_{\boldsymbol{b}',\boldsymbol{v}'})^2] = \|\boldsymbol{b}\boldsymbol{v}^\top - \boldsymbol{b}'(\boldsymbol{v}')^\top\|_F^2 - \|\boldsymbol{v} - \boldsymbol{v}'\|_F^2 - \|\boldsymbol{b} - \boldsymbol{b}'\|_F^2$$
$$= -2(1 - \boldsymbol{v}^\top\boldsymbol{v}')(1 - \boldsymbol{b}^\top\boldsymbol{b}') \leq 0.$$

Using Gordon's inequality, we get

$$\mathbb{E}[\inf_{\boldsymbol{v}} \sup_{\boldsymbol{b}} Z_{\boldsymbol{b},\boldsymbol{v}}] \leq \mathbb{E}[\inf_{\boldsymbol{v}} \sup_{\boldsymbol{b}} W_{\boldsymbol{b},\boldsymbol{v}}] = \mathscr{G}(\mathcal{B}) - \mathbb{E}[\|\boldsymbol{\xi}\|_2] \leq \mathscr{G}(\mathcal{B}) - \frac{n}{\sqrt{n+1}}.$$

To complete the proof of the first statement, it suffices to note that the mapping $\mathbf{Z} \mapsto \inf_{\boldsymbol{b}\in\mathcal{B}} \|\mathbf{Z}\boldsymbol{b}\|_2$ is Lipschitz with constant 1, and to apply the Gaussian concentration inequality (Boucheron et al., 2013, Theorem 5.6). Scaling the obtained bound by $1/\sqrt{n}$, the proof of the inequality in the second statement is immediate after we use the simple bound $(n/n+1)^{1/2} \geq 1 - 1/2n$. $\square$

**Lemma 4.** *Let $\mathbf{Z}$ be a $n \times p$ matrix with iid $\mathcal{N}(0,1)$ entries. Let $V$ be any compact subset of $\mathbb{S}^{p-1} \times \mathbb{S}^{n-1}$ and define $V_1 = \{\boldsymbol{v} : \exists \boldsymbol{u} \text{ s.t. } (\boldsymbol{v},\boldsymbol{u}) \in V\}$ and $V_2 = \{\boldsymbol{u} : \exists \boldsymbol{v} \text{ s.t. } (\boldsymbol{v},\boldsymbol{u}) \in V\}$. Then for any $n \geq 1$ and $t > 0$, with probability at least $1 - \exp(-t^2/2)$, we have*

$$\sup_{[\boldsymbol{v};\boldsymbol{u}]\in V} \boldsymbol{u}^\top\mathbf{Z}\boldsymbol{v} \leq \mathscr{G}(V_1) + \mathscr{G}(V_2) + t.$$

*Proof.* For each $(\boldsymbol{v},\boldsymbol{u}) \in V$, we define

$$Z_{\boldsymbol{v},\boldsymbol{u}} := \boldsymbol{u}^\top\mathbf{Z}\boldsymbol{v}, \qquad W_{\boldsymbol{v},\boldsymbol{u}} := \boldsymbol{v}^\top\boldsymbol{\xi} + \boldsymbol{u}^\top\bar{\boldsymbol{\xi}},$$

where $\boldsymbol{\xi}$ and $\bar{\boldsymbol{\xi}}$ are two independent standard Gaussian vectors. Therefore, $(\boldsymbol{v},\boldsymbol{u}) \mapsto Z_{\boldsymbol{v},\boldsymbol{u}}$ and $(\boldsymbol{v},\boldsymbol{u}) \mapsto W_{\boldsymbol{v},\boldsymbol{u}}$ define centered continuous Gaussian processes $W$ and $Z$ indexed by $V$.

To compute the variance of the increments of $W$. We remark that

$$Z_{\boldsymbol{v},\boldsymbol{u}} - Z_{\boldsymbol{v}',\boldsymbol{u}'} = \text{trace}[\mathbf{Z}(\boldsymbol{v}\boldsymbol{u}^\top - \boldsymbol{v}'(\boldsymbol{u}')^\top)] \sim \mathcal{N}(0, \|\boldsymbol{v}\boldsymbol{u}^\top - \boldsymbol{v}'(\boldsymbol{u}'^\top)\|_F^2).$$

Hence,

$$\mathbb{E}\big[\big(Z_{\boldsymbol{v},\boldsymbol{u}} - Z_{\boldsymbol{v}',\boldsymbol{u}'}\big)^2\big] = \|\boldsymbol{v}\boldsymbol{u}^\top - \boldsymbol{v}'(\boldsymbol{u}')^\top\|_F^2 = \|(\boldsymbol{v} - \boldsymbol{v}')\boldsymbol{u}^\top + \boldsymbol{v}'(\boldsymbol{u} - \boldsymbol{u}')^\top\|_F^2$$
$$\leq \|\boldsymbol{v} - \boldsymbol{v}'\|_2^2 + \|\boldsymbol{u} - \boldsymbol{u}'\|_2^2, \tag{29}$$

using Cauchy-Schwarz's inequality and the facts that $\boldsymbol{v}, \boldsymbol{v}' \in \mathbb{S}^{p-1}$ and $\boldsymbol{u}, \boldsymbol{u}' \in \mathbb{S}^{n-1}$. On the other hand, the definition of the process $Z$ yields

$$\mathbb{E}[(W_{\boldsymbol{v},\boldsymbol{u}} - W_{\boldsymbol{v}',\boldsymbol{u}'})^2] = \|\boldsymbol{v} - \boldsymbol{v}'\|_2^2 + \|\boldsymbol{u} - \boldsymbol{u}'\|_2^2. \tag{30}$$

From (29),(30), we conclude that the centered Gaussian processes $W$ and $Z$ satisfy the conditions of Gordon's inequality. Hence, using the notation $V_1 = \{\boldsymbol{v} : \exists \boldsymbol{u} \text{ s.t. } (\boldsymbol{v}, \boldsymbol{u}) \in V\}$ and $V_2 = \{\boldsymbol{u} : \exists \boldsymbol{v} \text{ s.t. } (\boldsymbol{v}, \boldsymbol{u}) \in V\}$, we get

$$\mathbb{E}\left[\sup_{[\boldsymbol{v};\boldsymbol{u}]\in V} Z_{\boldsymbol{v},\boldsymbol{u}}\right] \leq \mathbb{E}\left[\sup_{[\boldsymbol{v};\boldsymbol{u}]\in V} W_{\boldsymbol{v},\boldsymbol{u}}\right] \leq \mathbb{E}\left[\sup_{\boldsymbol{v}\in V_1} \boldsymbol{v}^\top\boldsymbol{\xi}\right] + \mathbb{E}\left[\sup_{\boldsymbol{u}\in V_2} \boldsymbol{u}^\top\bar{\boldsymbol{\xi}}\right] = \mathscr{G}(V_1) + \mathscr{G}(V_2).$$

Moreover, $\mathbf{Z} \mapsto \sup_{[\boldsymbol{v};\boldsymbol{u}]\in V_1 \times V_2} \boldsymbol{u}^\top \mathbf{Z}\boldsymbol{v}$ is Lipschitz continuous with constant 1, so the Gaussian concentration inequality holds (Boucheron et al., 2013, Theorem 5.6). This and the previous inequality bounding the mean complete the proof. $\square$

## 9.2 Removing compactness constraints: peeling techniques

**Lemma 5** (Single-parameter peeling). *Let $g : \mathbb{R}_+ \to \mathbb{R}_+$ be a right-continuous non-decreasing function and $h : V \to \mathbb{R}_+$. Assume that for some constants $b \in \mathbb{R}_+$ and $c \geq 1$, for every $r > 0$ and for any $\delta \in (0, 1/(7 \vee c))$, we have*

$$A(r, \delta) = \left\{ \inf_{\boldsymbol{v}\in V : h(\boldsymbol{v})\leq r} M(\boldsymbol{v}) \geq -g(r) - b\sqrt{\log(1/\delta)} \right\},$$

*with probability at least $1 - c\delta$. Then, with probability at least $1 - c\delta$, we have*

$$\forall \boldsymbol{v} \in V \quad M(\boldsymbol{v}) \geq -1.2(g \circ h)(\boldsymbol{v}) - \big(3 + \sqrt{\log(9/\delta)}\big)b.$$

*Proof.* Throughout the proof, without loss of generality, we assume $b = 1$. Let $\eta, \epsilon > 1$ be two parameters to be chosen later on. We set[5] $\mu_0 = 0$, $\mu_k = \mu\eta^{k-1}$, $\nu_k = g^{-1}(\mu_k)$ and $V_k = \{\boldsymbol{v} \in V : \mu_k \leq (g \circ h)(\boldsymbol{v}) < \mu_{k+1}\}$, for $k \geq 1$. The union bound and the fact that $\sum_{k\geq 1} k^{-1-\epsilon} \leq 1 + \epsilon^{-1}$ imply that the event

$$A := \bigcap_{k=1}^{\infty} A(\nu_k, \epsilon\delta/((1+\epsilon)k^{1+\epsilon}))$$

has a probability at least $1 - c\delta$. We assume in the sequel that this event is realized, that is

$$\forall k \in \mathbb{N}^* \quad \begin{cases} \forall \boldsymbol{v} \in V \text{ such that } h(\boldsymbol{v}) \leq \nu_k \text{ we have} \\ M(\boldsymbol{v}) \geq -g(\nu_k) - \sqrt{\log\{(1+\epsilon)/(\epsilon\delta)\} + (1+\epsilon)\log k}, \end{cases} \quad . \tag{31}$$

For every $\boldsymbol{v} \in V$, there is $\ell \in \mathbb{N}$ such that $\boldsymbol{v} \in V_\ell$. If $\ell \geq 1$, then $h(\boldsymbol{v}) \leq \nu_{\ell+1}$ and (31) implies that

$$M(\boldsymbol{v}) \geq -g(\nu_{\ell+1}) - \sqrt{\log\{(1+\epsilon)/(\epsilon\delta)\} + (1+\epsilon)\log(\ell+1)}$$
$$= -\mu_{\ell+1} - \sqrt{\log\{(1+\epsilon)/(\epsilon\delta)\} + (1+\epsilon)\log(\ell+1)}$$
$$= -\eta\mu_\ell - \sqrt{\log\{(1+\epsilon)/(\epsilon\delta)\} + (1+\epsilon)\log(\ell+1)}$$
$$\geq -\eta^2(g \circ h)(\boldsymbol{v}) + (\eta-1)\mu\eta^\ell - \sqrt{\log\{(1+\epsilon)/(\epsilon\delta)\} + (1+\epsilon)\log(\ell+1)}. \tag{32}$$

If $\ell = 0$, then (31) with $k = 1$ leads to

$$M(\boldsymbol{v}) \geq -g(\nu_1) - \sqrt{\log\{(1+\epsilon)/(\epsilon\delta)\}}$$
$$= -g(g^{-1}(\mu)) - \sqrt{\log\{(1+\epsilon)/(\epsilon\delta)\}}$$
$$= -\mu - \sqrt{\log\{(1+\epsilon)/(\epsilon\delta)\}}. \tag{33}$$

591  From (32) one can infer that, for $\ell \geq 1$,

$$M(\boldsymbol{v}) \geq -\eta^2 (g \circ h)(\boldsymbol{v}) - \sqrt{\log\{(1+\epsilon)/(\epsilon\delta)\}}$$
$$+ \eta^\ell \left( (\eta-1)\mu - \sup_{z \geq 1} \frac{\sqrt{\log\{(1+\epsilon)/(\epsilon\delta)\} + (1+\epsilon)\log(z+1)} - \sqrt{\log\{(1+\epsilon)/(\epsilon\delta)\}}}{\eta^z} \right).$$

592  We choose $\mu$ so that the last term vanishes, that is

$$(\eta-1)\mu = \sup_{z \geq 1} \frac{\sqrt{\log\{(1+\epsilon)/(\epsilon\delta)\} + (1+\epsilon)\log(z+1)} - \sqrt{\log\{(1+\epsilon)/(\epsilon\delta)\}}}{\eta^z}$$
$$= \sup_{z \geq 1} \frac{(1+\epsilon)\eta^{-z}\log(z+1)}{\sqrt{\log\{(1+\epsilon)/(\epsilon\delta)\} + (1+\epsilon)\log(z+1)} + \sqrt{\log\{(1+\epsilon)/(\epsilon\delta)\}}}.$$

593  To compute the last expression, we choose $\eta^2 = 1.2$ and $\epsilon = 1/8$. This yields

$$\mu = (\eta-1)^{-1} \sup_{z \geq 1} \frac{(9/8)(1.2)^{-z/2}\log(z+1)}{\sqrt{\log(9/\delta) + (9/8)\log(z+1)} + \sqrt{\log(9/\delta)}}$$
$$\leq (\eta-1)^{-1} \sup_{z \geq 1} \frac{(9/8)(1.2)^{-z/2}\log(z+1)}{\sqrt{\log 36 + (9/8)\log(z+1)} + \sqrt{\log 36}} \leq 3.$$

594  Combining with (33), this yields

$$M(\boldsymbol{v}) \geq -\mu - 1.2(g \circ h)(\boldsymbol{v}) - \sqrt{\log(9/\delta)}$$
$$\geq -1.2(g \circ h)(\boldsymbol{v}) - \left(3 + \sqrt{\log(9/\delta)}\right).$$

595  This completes the proof. $\qquad\square$

**Lemma 6** (Bi-parameter peeling). *Let $g, \bar{g}$ be right-continuous, non-decreasing functions from $\mathbb{R}_+$*
597  *to $\mathbb{R}_+$ and $h, \bar{h}$ be functions from $V$ to $\mathbb{R}_+$. Assume that for some constants $b \in \mathbb{R}_+$ and $c \geq 1$, for*
598  *every $r, \bar{r} > 0$ and for any $\delta \in (0, 1/(c \vee 7))$, we have*

$$A(r, \bar{r}, \delta) = \left\{ \inf_{\boldsymbol{v} \in V : (h,\bar{h})(\boldsymbol{v}) \leq (r,\bar{r})} M(\boldsymbol{v}) \geq -g(r) - \bar{g}(\bar{r}) - b\sqrt{\log(1/\delta)} \right\},$$

599  *with probability at least $1 - c\delta$. Then, with probability at least $1 - c\delta$, we have*

$$\forall \boldsymbol{v} \in V \quad M(\boldsymbol{v}) \geq -1.2(g \circ h)(\boldsymbol{v}) - 1.2(\bar{g} \circ \bar{h})(\boldsymbol{v}) - b\left(4.8 + \sqrt{\log(81/\delta)}\right).$$

*Proof.* We will repeat the same steps as for the one-parameter peeling. W.l.o.g. we assume $b = 1$. We
choose $\mu > 0$, $\eta > 1$ and $\epsilon > 0$. Define [6] $\mu_0 = 0$, $\mu_k = \mu\eta^{k-1}$, $\nu_k = g^{-1}(\mu_k)$, $\bar{\nu}_k = \bar{g}^{-1}(\mu_k)$ and
$V_{k,\bar{k}} = \{ \boldsymbol{v} \in V : \mu_k \leq (g \circ h)(\boldsymbol{v}) < \mu_{k+1}, \; \mu_{\bar{k}} \leq (\bar{g} \circ \bar{h})(\boldsymbol{v}) < \mu_{\bar{k}+1} \}$. The union bound implies
that the event

$$A = \bigcap_{k=1}^{\infty} A\left(\nu_k, \bar{\nu}_{\bar{k}}, \frac{\epsilon^2 \delta}{(1+\epsilon)^2 (k\bar{k})^{1+\epsilon}}\right)$$

600  has a probability at least $1 - c\delta$. To ease notation, set $\delta_\epsilon = \epsilon^2\delta/(1+\epsilon)^2$. We assume in the sequel
601  that the event $A$ is realized, that is

$$\forall k, \bar{k} \in \mathbb{N}^*, \; \forall \boldsymbol{v} \in V \text{ such that } (h, \bar{h})(\boldsymbol{v}) \leq (\nu_k, \bar{\nu}_{\bar{k}}) \text{ we have}$$
$$M(\boldsymbol{v}) \geq -g(\nu_k) - \bar{g}(\bar{\nu}_{\bar{k}}) - \sqrt{\log(1/\delta_\epsilon) + (1+\epsilon)\log(k\bar{k})}. \qquad (34)$$

602  For every $\boldsymbol{v} \in V$, there is a pair $(\ell, \bar{\ell}) \in \mathbb{N}^2$ such that $\boldsymbol{v} \in V_\ell$. If $\ell \wedge \bar{\ell} \geq 1$, then $(h, \bar{h})(\boldsymbol{v}) \leq$
603  $(\nu_{\ell+1}, \bar{\nu}_{\bar{\ell}+1})$, and (34) implies that

$$M(\boldsymbol{v}) \geq -g(\nu_{\ell+1}) - \bar{g}(\bar{\nu}_{\bar{\ell}+1}) - \sqrt{\log(1/\delta_\epsilon) + (1+\epsilon)\log(\ell+1)(\bar{\ell}+1)}$$
$$= -\mu_{\ell+1} - \mu_{\bar{\ell}+1} - \sqrt{\log(1/\delta_\epsilon) + (1+\epsilon)\log(\ell+1)(\bar{\ell}+1)}$$
$$= -\eta\mu_\ell - \eta\mu_{\bar{\ell}} - \sqrt{\log(1/\delta_\epsilon) + (1+\epsilon)\log(\ell+1)(\bar{\ell}+1)}.$$

From this inequality, we infer that

$$
\begin{aligned}
M(\boldsymbol{v}) \geq\ & -\eta^2[(g \circ h)(\boldsymbol{v}) + (\bar{g} \circ \bar{h})(\boldsymbol{v})] \\
& + \eta(\eta - 1)(\mu_\ell + \mu_{\bar\ell}) - \sqrt{\log(1/\delta_\epsilon) + (1+\epsilon)\log(\ell+1)(\bar\ell+1)} \\
=\ & -\eta^2(g \circ h)(\boldsymbol{v}) - \eta^2(\bar{g} \circ \bar{h})(\boldsymbol{v}) - \sqrt{\log(1/\delta_\epsilon)} \\
& + \left\{ (\eta-1)\mu(\eta^\ell + \eta^{\bar\ell}) + \sqrt{\log(1/\delta_\epsilon)} - \sqrt{\log(1/\delta_\epsilon) + (1+\epsilon)\log(\ell+1)(\bar\ell+1)} \right\}.
\end{aligned}
$$

We choose $\mu$ so that the expression inside the braces is nonnegative, that is

$$
(\eta-1)\mu = \sup_{z,\bar{z} \geq 1} \frac{\sqrt{\log(1/\delta_\epsilon) + (1+\epsilon)\log(1+z) + (1+\epsilon)\log(1+\bar z)} - \sqrt{\log(1/\delta_\epsilon)}}{\eta^z + \eta^{\bar z}}.
$$

Setting $\epsilon = 1/8$, $\eta^2 = 1.2$ and using that $\delta \leq 1/7$, we get that $\delta_\epsilon \leq 1/567$ and hence

$$
\mu \leq (\eta-1)^{-1} \sup_{z,\bar{z} \geq 1} \frac{\sqrt{\log 567 + (9/8)\log(1+z) + (9/8)\log(1+\bar z)} - \sqrt{\log 567}}{1.2^{z/2} + 1.2^{\bar z/2}} \leq 2.4
$$

Combining with the case $\ell \wedge \bar\ell = 1$, this yields

$$
\begin{aligned}
M(\boldsymbol{v}) &\geq -2\mu - 1.2(g \circ h)(\boldsymbol{v}) - 1.2(\bar g \circ \bar h)(\boldsymbol{v}) - \sqrt{\log(81/\delta)} \\
&\geq -1.2(g \circ h)(\boldsymbol{v}) - 1.2(\bar g \circ \bar h)(\boldsymbol{v}) - 4.8 - \sqrt{\log(81/\delta)}.
\end{aligned}
$$

This completes the proof. $\qquad\qquad\square$

## 9.3 Structural properties of Gaussian designs

**Proposition 3.** *Let $\mathbf{Z}$ be a $n \times p$ matrix with iid $\mathcal{N}_p(0, \boldsymbol{\Sigma})$ columns. For all $n \geq 100$ and $\delta \in (0, 1/7]$, with probability at least $1 - \delta$, the following inequality holds: for all $\boldsymbol{v} \in \mathbb{R}^p$,*

$$
\left\| \mathbf{Z}^{(n)}\boldsymbol{v} \right\|_2 \geq \left( 1 - \frac{4.3 + \sqrt{2\log(9/\delta)}}{\sqrt{n}} \right) \left\| \boldsymbol{\Sigma}^{1/2}\boldsymbol{v} \right\|_2 - \frac{1.2\mathscr{G}(\boldsymbol{\Sigma}^{1/2}\mathbb{B}_1^p)}{\sqrt{n}} \|\boldsymbol{v}\|_1. \tag{35}
$$

**Remark 1.** The above result is similar to (Raskutti et al., 2010, Theorem 1), but it has three advantages. First, the influence of the failure probability $\delta$ on the constants is made explicit. Second, the factor $\rho(\boldsymbol{\Sigma})$ appearing in the last term is replaced by the smaller quantity $\mathscr{G}(\boldsymbol{\Sigma}^{1/2}\mathbb{B}_1^p)$. Third, we improved the constants.

Proposition 3 is a useful technical tool that allows one to transfer the restricted eigenvalue property from the population covariance matrix to the empirical one. Following Oliveira (2013) we refer to (35) as the transfer principle.

*Proof of Proposition 3.* Let $r > 0$. We define define the sets

$$
V_{\boldsymbol{\Sigma}}(r) := \{ \boldsymbol{v} \in \mathbb{R}^p : \|\boldsymbol{\Sigma}^{1/2}\boldsymbol{v}\|_2 = 1, \|\boldsymbol{v}\|_1 \leq r \},
$$

and $\mathcal{B} := \{ \boldsymbol{\Sigma}^{1/2}\boldsymbol{v} : \boldsymbol{v} \in V_{\boldsymbol{\Sigma}}(r) \}$. Note that, if $\boldsymbol{\xi} \sim \mathcal{N}_p(0, \mathbf{I}_p)$,

$$
\mathscr{G}(\mathcal{B}) \leq \mathbb{E}\left[ \sup_{\boldsymbol{v} \in r\mathbb{B}_1^p} \boldsymbol{\xi}^\top \boldsymbol{\Sigma}^{1/2}\boldsymbol{v} \right] \leq r\mathscr{G}(\boldsymbol{\Sigma}^{1/2}\mathbb{B}_1^p). \tag{36}
$$

Let $\widetilde{\mathbf{Z}}$ be a $n \times p$ matrix with iid $\mathcal{N}(0,1)$ entries such that $\mathbf{Z} = \widetilde{\mathbf{Z}}\boldsymbol{\Sigma}^{1/2}$. Clearly,

$$
\inf_{\boldsymbol{v} \in V_{\boldsymbol{\Sigma}}(r)} \left\| \mathbf{Z}^{(n)}\boldsymbol{v} \right\|_2 = \inf_{\boldsymbol{b} \in \mathcal{B}} \left\| \widetilde{\mathbf{Z}}^{(n)}\boldsymbol{b} \right\|_2.
$$

The above equality, (36) and Lemma 4 (noting that $\mathcal{B} \subset \mathbb{S}^{p-1}$) entails that, for all $r > 0$ and $\delta \in (0, 1]$, with probability at least $1 - \delta$, the following inequality holds:

$$
\inf_{\boldsymbol{v} \in V_{\boldsymbol{\Sigma}}(r)} \left\| \mathbf{Z}^{(n)}\boldsymbol{v} \right\|_2 \geq 1 - \frac{1}{2n} - \sqrt{\frac{2\log(1/\delta)}{n}} - \frac{\mathscr{G}(\boldsymbol{\Sigma}^{1/2}\mathbb{B}_1^p)}{\sqrt{n}}r.
$$

We will now use the above property and Lemma 5 with constraint set $V := \{\boldsymbol{v} \in \mathbb{R}^p : \|\boldsymbol{\Sigma}^{1/2}\boldsymbol{v}\|_2 = 1\}$,

$$M(\boldsymbol{v}) := \left\|\mathbf{Z}^{(n)}\boldsymbol{v}\right\|_2 - 1 + \frac{1}{2n},$$

functions $h(\boldsymbol{v}) := \|\boldsymbol{v}\|_1$, $g(r) := \frac{\mathscr{G}(\boldsymbol{\Sigma}^{1/2}\mathbb{B}_1^p)}{\sqrt{n}}r$, and constants $c := 1$ and $b := \sqrt{2/n}$. Lemma 5 implies that with probability at least $1 - \delta$, for all $\boldsymbol{v}$ such that $\|\boldsymbol{\Sigma}^{1/2}\boldsymbol{v}\|_2 = 1$, we have

$$M(\boldsymbol{v}) = \left\|\mathbf{Z}^{(n)}\boldsymbol{v}\right\|_2 - 1 + \frac{1}{2n} \geq -1.2\frac{\mathscr{G}(\boldsymbol{\Sigma}^{1/2}\mathbb{B}_1^p)}{\sqrt{n}}\|\boldsymbol{v}\|_1 - \frac{3\sqrt{2} + \sqrt{2\log(9/\delta)}}{\sqrt{n}}.$$

Replacing $\boldsymbol{v}$ by $\boldsymbol{u}/\|\boldsymbol{\Sigma}^{1/2}\boldsymbol{u}\|_2$, for an arbitrary $\boldsymbol{u} \in \mathbb{R}^p$, we get

$$\left\|\mathbf{Z}^{(n)}\boldsymbol{u}\right\|_2 \geq \left(1 - \frac{1}{2n} - \frac{3\sqrt{2} + \sqrt{2\log(9/\delta)}}{\sqrt{n}}\right)\|\boldsymbol{\Sigma}^{1/2}\boldsymbol{u}\| - 1.2\frac{\mathscr{G}(\boldsymbol{\Sigma}^{1/2}\mathbb{B}_1^p)}{\sqrt{n}}\|\boldsymbol{u}\|_1.$$

To complete the proof, it suffices to note that $(1/2\sqrt{n}) + 3\sqrt{2} \leq 4.3$ for $n \geq 100$. $\qquad\square$

**Proposition 4.** *Let $\mathbf{Z} \in \mathbb{R}^{n \times p}$ be a random matrix with i.i.d. $\mathcal{N}_p(0, \boldsymbol{\Sigma})$ rows. For all $\delta \in (0, 1]$ and $n \in \mathbb{N}$, with probability at least $1 - \delta$, the following property holds: for all $[\boldsymbol{v}; \boldsymbol{u}] \in \mathbb{R}^{p+n}$,*

$$\left|\boldsymbol{u}^\top \mathbf{Z}^{(n)}\boldsymbol{v}\right| \leq \|\boldsymbol{\Sigma}^{1/2}\boldsymbol{v}\|_2\|\boldsymbol{u}\|_2\sqrt{\frac{2}{n}}\left(4.8 + \sqrt{\log(81/\delta)}\right)$$

$$+ 1.2\|\boldsymbol{v}\|_1\|\boldsymbol{u}\|_2\frac{\mathscr{G}(\boldsymbol{\Sigma}^{1/2}\mathbb{B}_1^p)}{\sqrt{n}} + 1.2\|\boldsymbol{\Sigma}^{1/2}\boldsymbol{v}\|_2\frac{\mathscr{G}(\|\boldsymbol{u}\|_1\mathbb{B}_1^n \cap \|\boldsymbol{u}\|_2\mathbb{B}_2^n)}{\sqrt{n}}.$$

**Remark 2.** If, instead of Proposition 4, well-known upper bounds on the maximal singular value of a Gaussian matrix, we get a sub-optimal result. Indeed, upper tail bounds on largest singular value imply that, with high-probability, for all $\boldsymbol{v}$ and $\boldsymbol{u}$,

$$\left|\boldsymbol{u}^\top \mathbf{Z}^{(n)}\boldsymbol{v}\right| \leq \|\boldsymbol{\Sigma}^{1/2}\boldsymbol{v}\|_2\|\boldsymbol{u}\|_2\|\mathbf{Z}^{(n)}\boldsymbol{\Sigma}^{-1/2}\|_{op} \lesssim \|\boldsymbol{\Sigma}^{1/2}\boldsymbol{v}\|_2\|\boldsymbol{u}\|_2\sqrt{\frac{p}{n}}.$$

In case $\boldsymbol{v}$ and $\boldsymbol{u}$ are sparse, the previous lemma establishes a much sharp upper bound with respect to dimension. One may see Proposition 4 also as generalized control on the "incoherence" between the column-space of $\mathbf{Z}^{(n)}$ and the identity $\mathbf{I}_n$. This is particularly useful when the vectors are sparse as in our setting. Alongside Proposition 3, Proposition 4 is at the core of our methodology to obtain improved near-optimal rates for corrupted sparse linear regression.

*Proof.* Let $r_1, r_2 > 0$ and define the sets

$$V_{\boldsymbol{\Sigma}, 1}(r_1) := \{\boldsymbol{v} \in \mathbb{R}^p : \|\boldsymbol{\Sigma}^{1/2}\boldsymbol{v}\|_2 = 1, \|\boldsymbol{v}\|_1 \leq r_1\},$$
$$V_2(r_2) := \{\boldsymbol{u} \in \mathbb{R}^n : \|\boldsymbol{u}\|_2 = 1, \|\boldsymbol{u}\|_1 \leq r_2\}.$$

We also define the set $\mathcal{B}_1 := \{\boldsymbol{\Sigma}^{1/2}\boldsymbol{v} : \boldsymbol{v} \in V_{\boldsymbol{\Sigma}, 1}(r_1)\}$. By similar arguments used to establish (36), we have the following Gaussian width bounds:

$$\mathscr{G}(\mathcal{B}_1) \leq r_1\mathscr{G}(\boldsymbol{\Sigma}^{1/2}\mathbb{B}_1^p), \quad \mathscr{G}(V_2(r_2)) \leq r_2\mathscr{G}(\mathbb{B}_1^n \cap \mathbb{B}_2^n/r_2). \tag{37}$$

Let $\widetilde{\mathbf{Z}}$ be a $n \times p$ matrix with iid $\mathcal{N}(0, 1)$ entries such that $\mathbf{Z} = \widetilde{\mathbf{Z}}\boldsymbol{\Sigma}^{1/2}$. Clearly,

$$\sup_{[\boldsymbol{v}; \boldsymbol{u}] \in V_{\boldsymbol{\Sigma}, 1}(r_1) \times V_2(r_2)} |\boldsymbol{u}^\top \mathbf{Z}^{(n)}\boldsymbol{v}| = \sup_{[\boldsymbol{v}'; \boldsymbol{u}] \in \mathcal{B}_1 \times V_2(r_2)} |\boldsymbol{u}^\top \widetilde{\mathbf{Z}}^{(n)}\boldsymbol{v}'|.$$

The above equality, (37) and Lemma 4 (noting that $\mathcal{B}_1 \subset \mathbb{S}^{p-1}$ and $V_2(r_2) \subset \mathbb{S}^{n-1}$) entail that, for any $r_1, r_2 > 0$ and $\delta \in (0, 1]$, with probability at least $1 - \delta$, the following inequality holds:

$$\sup_{[\boldsymbol{v}; \boldsymbol{u}] \in V_{\boldsymbol{\Sigma}, 1}(r_1) \times V_2(r_2)} |\boldsymbol{u}^\top \mathbf{Z}^{(n)}\boldsymbol{v}| \leq \frac{\mathscr{G}(\boldsymbol{\Sigma}^{1/2}\mathbb{B}_1^p)}{\sqrt{n}}r_1 + \frac{\mathscr{G}(\mathbb{B}_1^n \cap \mathbb{B}_2^n/r_2)}{\sqrt{n}}r_2 + \sqrt{\frac{2\log(1/\delta)}{n}}.$$

We use the above property and Lemma 6 with constraint sets $V_1 := \{\boldsymbol{v} \in \mathbb{R}^p : \|\boldsymbol{\Sigma}^{1/2}\boldsymbol{v}\|_2 = 1\}$ and $V_2 := \{\boldsymbol{u} \in \mathbb{R}^n : \|\boldsymbol{v}\|_2 = 1\}$, functions $M(\boldsymbol{u}) := |\boldsymbol{u}^\top \mathbf{Z}^{(n)}\boldsymbol{v}|$ and

$$h(\boldsymbol{v}) := \|\boldsymbol{v}\|_1, \qquad \bar{h}(\boldsymbol{u}) := \|\boldsymbol{u}\|_1, \qquad g(r_1) := \frac{\mathscr{G}(\boldsymbol{\Sigma}^{1/2}\mathbb{B}_1^p)}{\sqrt{n}}\, r_1, \quad \bar{g}(r_2) := \frac{\mathscr{G}(\mathbb{B}_1^n \cap \mathbb{B}_2^n/r_2)}{\sqrt{n}}\, r_2,$$

and constants $c := 1$ and $b := \sqrt{2/n}$. The desired inequality follows from Lemma 6 combined with the fact that

$$\left[ \frac{\boldsymbol{v}}{\|\boldsymbol{\Sigma}^{1/2}\boldsymbol{v}\|_2} ; \frac{\boldsymbol{u}}{\|\boldsymbol{u}\|_2} \right] \in V_{\boldsymbol{\Sigma},1}(r_1) \times V_2(r_2),$$

for all $[\boldsymbol{v};\boldsymbol{u}] \in \mathbb{R}^p \times \mathbb{R}^n$ and the homogeneity of norms. $\qquad\square$

**Lemma 7** ($\mathrm{TP}_{\boldsymbol{\Sigma}} + \mathrm{IP}_{\boldsymbol{\Sigma}} \Rightarrow \mathrm{ATP}_{\boldsymbol{\Sigma}}$). *Let $\mathbf{Z} \in \mathbb{R}^{n \times p}$ be a matrix satisfying $\mathrm{TP}_{\boldsymbol{\Sigma}}(\mathsf{a}_1;\mathsf{a}_2)$ and $\mathrm{IP}_{\boldsymbol{\Sigma}}(\mathsf{b}_1;\mathsf{b}_2;\mathsf{b}_3)$ for some positive numbers $\mathsf{a}_1$, $\mathsf{a}_2$, $\mathsf{b}_1$, $\mathsf{b}_2$ and $\mathsf{b}_3$. Then, for any $\alpha > 0$, $\mathbf{Z}$ satisfies the $\mathrm{ATP}_{\boldsymbol{\Sigma}}(\mathsf{c}_1;\mathsf{c}_2;\mathsf{c}_3)$ with constants $\mathsf{c}_1 = \sqrt{\mathsf{a}_1^2 - \mathsf{b}_1 - \alpha^2}$, $\mathsf{c}_2 = \mathsf{a}_2 + \mathsf{b}_2/\alpha$ and $\mathsf{c}_3 = \mathsf{b}_3/\alpha$. Taking $\alpha = \mathsf{a}_1/2$, we obtain that $\mathrm{ATP}_{\boldsymbol{\Sigma}}(\mathsf{c}_1;\mathsf{c}_2;\mathsf{c}_3)$ holds with constants $\mathsf{c}_1 = \sqrt{(3/4)\mathsf{a}_1^2 - \mathsf{b}_1 - \alpha^2}$, $\mathsf{c}_2 = \mathsf{a}_2 + 2\mathsf{b}_2/\mathsf{a}_1$ and $\mathsf{c}_3 = 2\mathsf{b}_3/\mathsf{a}_1$.*

*Proof.* Simple algebra and the TP property entail

$$\mathsf{c}_1\left\{ \|\boldsymbol{\Sigma}^{1/2}\boldsymbol{v}\|_2^2 + \|\boldsymbol{u}\|_2^2 \right\}^{1/2} = \left\{ \mathsf{a}_1^2\|\boldsymbol{\Sigma}^{1/2}\boldsymbol{v}\|_2^2 + \mathsf{a}_1^2\|\boldsymbol{u}\|_2^2 - (\mathsf{b}_1 + \alpha^2)(\|\boldsymbol{\Sigma}^{1/2}\boldsymbol{v}\|_2^2 + \|\boldsymbol{u}\|_2^2) \right\}^{1/2}$$

$$\overset{\mathrm{TP}_{\boldsymbol{\Sigma}}}{\leq} \left\{ \left(\|\mathbf{Z}^{(n)}\boldsymbol{v}\|_2 + \mathsf{a}_2\|\boldsymbol{v}\|_1\right)^2 + \mathsf{a}_1^2\|\boldsymbol{u}\|_2^2 - (\mathsf{b}_1 + \alpha^2)(\|\boldsymbol{\Sigma}^{1/2}\boldsymbol{v}\|_2^2 + \|\boldsymbol{u}\|_2^2) \right\}^{1/2}$$

$$\leq \left\{ \|\mathbf{Z}^{(n)}\boldsymbol{v}\|_2^2 + \|\boldsymbol{u}\|_2^2 - (\mathsf{b}_1 + \alpha^2)(\|\boldsymbol{\Sigma}^{1/2}\boldsymbol{v}\|_2^2 + \|\boldsymbol{u}\|_2^2) \right\}^{1/2} + \mathsf{a}_2\|\boldsymbol{v}\|_1.$$

By Young's inequality and IP, we get

$$\|\mathbf{Z}^{(n)}\boldsymbol{v}\|_2^2 + \|\boldsymbol{u}\|_2^2 = \|\mathbf{Z}^{(n)}\boldsymbol{v} + \boldsymbol{u}\|_2^2 - 2\boldsymbol{u}^\top \mathbf{Z}^{(n)}\boldsymbol{v}$$

$$\overset{\mathrm{IP}_{\boldsymbol{\Sigma}}}{\leq} \|\mathbf{Z}^{(n)}\boldsymbol{v} + \boldsymbol{u}\|_2^2 + 2\mathsf{b}_1\|\boldsymbol{\Sigma}^{1/2}\boldsymbol{v}\|_2\|\boldsymbol{u}\|_2 + 2\mathsf{b}_2\|\boldsymbol{v}\|_1\|\boldsymbol{u}\|_2 + 2\mathsf{b}_3\|\boldsymbol{\Sigma}^{1/2}\boldsymbol{v}\|_2\|\boldsymbol{u}\|_1$$

$$\overset{\mathrm{Young}}{\leq} \|\mathbf{Z}^{(n)}\boldsymbol{v} + \boldsymbol{u}\|_2^2 + (\mathsf{b}_1 + \alpha^2)\left( \|\boldsymbol{\Sigma}^{1/2}\boldsymbol{v}\|_2^2 + \|\boldsymbol{u}\|_2^2 \right) + \frac{\mathsf{b}_2^2}{\alpha^2}\|\boldsymbol{v}\|_1^2 + \frac{\mathsf{b}_3^2}{\alpha^2}\|\boldsymbol{u}\|_1^2.$$

To get the claimed result, it suffices to put the previous two inequalities together and to rearrange the terms. $\qquad\square$

Proposition 3, Proposition 4 and Lemma 7 entail immediately that the $\mathrm{ATP}_{\boldsymbol{\Sigma}}$ holds with high-probability.

**Corollary 1** ($\mathrm{ATP}_{\boldsymbol{\Sigma}}$ property for correlated Gaussian designs). *Let $\mathbf{Z} \in \mathbb{R}^{n \times p}$ be a random matrix with iid $\mathcal{N}_p(0, \boldsymbol{\Sigma})$ rows. Suppose $\delta \in (0, 1/7]$, $n \geq 100$ and $\alpha > 0$ are such that*

$$C_{n,\delta} := \left( 1 - \frac{4.3 + \sqrt{2\log(9/\delta)}}{\sqrt{n}} \right)^2 - \sqrt{\frac{2}{n}}\left( 4.8 + \sqrt{\log(81/\delta)} \right) - \alpha^2 > 0.$$

*Then, with probability at least $1 - 2\delta$, the following property holds: for all $[\boldsymbol{v};\boldsymbol{u}] \in \mathbb{R}^{p+n}$,*

$$\|\mathbf{Z}^{(n)}\boldsymbol{v} + \boldsymbol{u}\|_2 \geq C_{n,\delta}^{1/2}\left\| [\boldsymbol{\Sigma}^{1/2}\boldsymbol{v};\boldsymbol{u}] \right\|_2 - 1.2\left(1 + \frac{1}{\alpha}\right)\frac{\mathscr{G}(\boldsymbol{\Sigma}^{1/2}\mathbb{B}_1^p)}{\sqrt{n}}\|\boldsymbol{v}\|_1 - \frac{1.2}{\alpha}\frac{\mathscr{G}(\|\boldsymbol{u}\|_1\mathbb{B}_1^n \cap \|\boldsymbol{u}\|_2\mathbb{B}_2^n)}{\sqrt{n}}.$$

**Remark 3.** The particular choice $\alpha = 1/2$, in conjunction with the bound (28) on the Gaussian width, leads to the simpler bound

$$\|\mathbf{Z}^{(n)}\boldsymbol{v} + \boldsymbol{u}\|_2 \geq C_{n,\delta}^{1/2}\left\| [\boldsymbol{\Sigma}^{1/2}\boldsymbol{v};\boldsymbol{u}] \right\|_2 - \frac{3.6\mathscr{G}(\boldsymbol{\Sigma}^{1/2}\mathbb{B}_1^p)}{\sqrt{n}}\|\boldsymbol{v}\|_1 - 2.4\sqrt{\frac{2\log n}{n}}\|\boldsymbol{u}\|_1$$

with

$$C_{n,\delta} = \frac{3}{4} - \frac{17.5 + 9.6\sqrt{2\log(2/\delta)}}{\sqrt{n}}$$

**Remark 4.** If the goal was to fight against logarithmic factors, we could use a tighter bound on the Gaussian width of a convex polytope (Bellec, 2017, Prop. 1). It allows us to replace the term $\sqrt{2\log n}\,\|\boldsymbol{u}\|_1$ by $4\sqrt{1\vee\log(8en\|\boldsymbol{u}\|_2^2/\|\boldsymbol{u}\|_1^2)}\,\|\boldsymbol{u}\|_1$. On the one hand, if $\|\boldsymbol{u}\|_1^2\geq(o/e)\|\boldsymbol{u}\|_2^2$, then

$$4\sqrt{1\vee\log(8en\|\boldsymbol{u}\|_2^2/\|\boldsymbol{u}\|_1^2)}\,\|\boldsymbol{u}\|_1\leq 4\sqrt{1\vee\log(8e^2n/o)}\,\|\boldsymbol{u}\|_1. \tag{38}$$

On the other hand, if $\|\boldsymbol{u}\|_1^2\leq o\|\boldsymbol{u}\|_2^2$, then we can use the fact that the function $x\mapsto x\sqrt{1\vee\log(e/x^2)}=:\varphi(x)$ is increasing, we get

$$
\begin{aligned}
4\sqrt{1\vee\log(8en\|\boldsymbol{u}\|_2^2/\|\boldsymbol{u}\|_1^2)}\,\|\boldsymbol{u}\|_1 &= 4\sqrt{8en}\,\|\boldsymbol{u}\|_2\varphi\Big(\frac{\|\boldsymbol{u}\|_1}{\sqrt{8n}\,\|\boldsymbol{u}\|_2}\Big)\\
&\leq 4\sqrt{8en}\,\|\boldsymbol{u}\|_2\varphi\big(\sqrt{o/8en}\big)\\
&= 4\sqrt{eo}\,\|\boldsymbol{u}\|_2\sqrt{1+\log(8n/o)}.
\end{aligned}
\tag{39}$$

Combining (38) and (39), we get

$$\mathscr{G}(\|\boldsymbol{u}\|_1\mathbb{B}_1^n\cap\|\boldsymbol{u}\|_2\mathbb{B}_2^n)\leq 4(\|\boldsymbol{u}\|_1+\sqrt{o}\,\|\boldsymbol{u}\|_2)\sqrt{2+\log(8n/o)}.$$

If the proportion $o/n$ is fixed, or tends to zero at a rate slower than polynomial in $n$, this latter bound can be used to remove logarithmic terms.

# 10 Propositions imply theorems

The three theorems stated in the main body of the paper are simple consequences of the propositions established in this supplementary material. The aim of this section is to quickly show how the theorems can be derived from the corresponding propositions.

**Proof of Theorem 1** Theorem 1 is essentially a simplified version of Proposition 2. First, note that condition on $\lambda$ in Theorem 1, combined with the well-known upper bounds on the tails of maxima of Gaussian random variables (Boucheron et al., 2013), implies that $\lambda$ satisfies condition (iv) of Proposition 2. Furthermore, under the conditions of the theorem, conditions (i)-(iii) of Proposition 2, as well as (21) and (22), are satisfied with $\gamma=1$, $\mathsf{a}_1=\mathsf{c}_1\leq 1$, $\mathsf{a}_2=\mathsf{c}_2$ and $\mathsf{b}_1=0$. Replacing all these values in the inequality of Proposition 2, we get the claim of Theorem 1.

**Proof of Theorem 2** From Proposition 3 and the fact that $\mathscr{G}(\boldsymbol{\Sigma}^{1/2}\mathbb{B}_1^p)\leq\sqrt{2\log p}$, we infer that the $\mathrm{TP}_{\boldsymbol{\Sigma}}$ is satisfied with appropriate constants $\mathsf{a}_1,\mathsf{a}_2$ with probability at least $1-\delta$. Similarly, Proposition 4 and the aforementioned bound on the Gaussian width imply that the $\mathrm{IP}_{\boldsymbol{\Sigma}}$ is satisfied with appropriate constants with probability at least $1-\delta$. In the intersection of these two events, according to Remark 3, $\mathrm{ATP}_{\boldsymbol{\Sigma}}$ is satisfied with $\mathsf{c}_1$, $\mathsf{c}_2$ and $\mathsf{c}_3$ as in the claim of Theorem 2.

**Proof of Theorem 3** Under the condition $\delta\geq 2e^{-\mathsf{d}_2 n}$, we check that $\mathsf{a}_1$ and $\mathsf{c}_1$ are constants. Therefore, combining the claims of Theorem 1, Theorem 2 and Lemma 2, we get the claim of Theorem 3.

## Footnotes

[5] Here $g^{-1}$ is the generalized inverse defined by $g^{-1}(x) = \inf\{a \in \mathbb{R}_+ : g(a) \geq x\}$.

[6]Here $g^{-1}$ is the generalized inverse given by $g^{-1}(x) = \inf\{a \in \mathbb{R}_+ : g(a) \geq x\}$.