[Reviews · NeurIPS 2019]

Reviewer 1



This is a very good paper. I really enjoyed reading the paper. The result is very strong and the paper is very readable. I strongly recommend accepting the paper, if the proof is correct. (I could not check the supplementary material due to time constraints.) This paper solves an important open problem in robust estimation. The open problem is described in the beginning: "Is it possible to attain optimal rates of estimation in outlier-robust sparse regression using penalized empirical risk minimization (PERM) with convex loss and convex penalties?" The answer was negative in the past researches, but the present paper gives the answer "YES". The estimate is very simple, because it is obtained from the minimization of the simple Huber-loss with the $l_1$ penalty with some devised tuning parameter $\lambda_o$, which is an interesting point. To show the optimal convergence rate, we need some additional assumptions that the authors honestly mention on l.26-27. However, they are not strong. The introduction is very clear. The authors give a clear history of the study of convergence rate in robust estimation. The key sentences are: "this result (which means the result obtained in this paper) is not valid in the most general situation, but we demonstrate its validity under the assumptions that the design matrix satisfies some incoherence condition and only the response is subject to contamination." and "The main goal of the present paper is to show that this sub-optimality is not an intrinsic property of the estimator (3), but rather an artefact of previous proof techniques. By using a refined argument, we prove that $\hat{\beta}$ defined by (3) does attain the optimal rate under very mild assumptions." Section 2 gives a key theorem. In particular, the authors illustrate why some complicated assumptions are necessary in the first paragraph with interesting examples, and they also give the main point of the proof which is to treat the extra parameter $\theta$ as a nuisance parameter when we obtain the bound, but not in the past. Section 3 focuses on the Gaussian design. Section 4 gives a very clear survey on prior work. The authors give future works very clearly in Section 5, including technical details amazingly. Section 6 is a numerical experiment, which verifies the order of $o/n$. This is a very small experiment, but this is enough, because the main purpose of this paper is a theoretical one. Major Comment I have only one comment. I think the largest feature is that the tuning parameter $\lambda_o$ is incorporated into the Huber loss in a different way. The usual $l_1$ penalized Huber loss function with $\lambda_o=1$ will not give the optimal convergence rate. What is the role of $\lambda_o$ on eq.(5)? (although the role is clear on eq.(3).) $\Phi(u)=(1/2)u^2 \cap (|u|-1/2)$. The effect of $\lambda_o$ vanishes for $u^2$, but it remains for (|u|-1/2), which gives the loss function $\lambda_o\sqrt{n} ( (1/n)\sum_{i=1}^n |y_i-X_i^\top \beta| -1/2 )$. When $\lambda_o$ has the order used in Theorem 3, the factor $\lambda_o\sqrt{n}$ has the order $(\log(n))^(1/2)$, which converges to infinity. This implies that the effect of large error of $|y_i-X_i^\top \beta|$ is not admitted at all. As a result, the main loss is the squared error only. Is this an appropriate point of view? If you have a clear point of view on the role of $\lambda_o$ on eq.(5), please give related comments. -------------------------------------------- Thank you for your reply.

Reviewer 2



I thank the authors for their detailed response. There is another NeuIPS submission whose results supersedes this one (this paper's result is only a corollary of that submission), and I will leave the decision to the AC. ===== - originality This paper only deals with the setting where response variables (y), and has to assume that x are sub-Gaussian, and similar results were already established in (Bhatia et al., 2017). However, (Liu et al., 2019) can handle corruptions in both x and y, although the bound is slightly worse. - quality The proofs are sound. - clarity The paper is well written. - significance There are many papers in this area, and this work has not differentiated itself from other papers.

Reviewer 3



I enjoyed reading this paper and support its publication in NeurIPS. The primary contribution of this work is technical and theoretical, but it provides a sharpened analysis of a very practical algorithm for robust regression. This work shows that under certain natural conditions, ell-1 penalized least-squares achieves the minimax rate of convergence for this problem, up to a logarithmic factor. I am not an expert in this area, but according to the authors, previous analyses of this same algorithm failed to show this sharp rate, and previous algorithms achieving this rate were complicated and difficult to use in practice. The technical innovation applies the KKT condition for beta, at the estimated outlier-contamination vector theta, to derive a recursive bound for the squared-error of the beta estimate. A main term of this bound is v'X'u for (u,v) the errors in (theta,beta), and a main technical insight is that this can be controlled by ||u||_2*||v||_1/sqrt(n) + ||u||_1*||v||_2/sqrt(n), rather than the naive operator-norm bound ||v||_2*||u||_2, when the design X satisfies an incoherence property with the standard basis. The improved bound then applies this insight and an a priori bound on (u,v) from a more standard Lasso-regression analysis. The authors demonstrate that the required incoherence property holds for (correlated) multivariate Gaussian designs, using a Gaussian-process and peeling argument. In my view, this insight is non-trivial, and both the paper and proof are also well-written. A few comments/minor typos: (1) I think some more explanation is needed after Definition 1, to explain how this captures the notions of restricted invertibility and incoherence in the previous paragraph. In particular, the role of the transfer principles in restricted invertibility is a bit confusing, as the RE condition for Sigma is only discussed later on the page. Explaining why (ii) captures some notion of incoherence would also be helpful. (2) It took me a while to find where X^{(n)} and \xi^{(n)} are defined---perhaps this can be clarified more explicitly. (3) Line 101, p-th largest should be p-th smallest (4) Line 476, J should be S and beta_j should be beta_j^* (5) Is there a minus sign missing in the first term on the right of Lemma 1, and its application in line (525)? (6) I'm not sure what trace means above line 564 for W_{b,v}. ------------- Post-rebuttal: Thanks to the authors for the response, clarification, and discussion.

[Author Response · NeurIPS 2019]

First, we would like to thank the reviewers for the careful reading of the paper and for helpful/thoughtful remarks. All the recommendations made by the reviewers will be taken into account in the revised version.

**Response to Reviewer # 1**   Thank you very much for your very enthusiastic report. Let us briefly comment on the question related to the choice of the parameter $\lambda_o$ and on its impact on the quality of estimation.

The basic intuition included in your report is correct: the fact that $\lambda_o\sqrt{n}$ tends to infinity when $n \to \infty$ implies that all the outliers for which the outlyingness $y_i - \boldsymbol{X}_i^\top\boldsymbol{\beta}^* - \xi_i$ is smaller than $c\sigma\sqrt{\log n}$ will be concerned only by the squared loss. As a consequence, if we were aware that all the outliers satisfy the condition $|y_i - \boldsymbol{X}_i^\top\boldsymbol{\beta}^* - \xi_i| \le c\sigma\sqrt{\log n}$ for some $c > 0$, there would be no need of using the Huber loss; the standard (penalized) least-squares estimator would have the statistical precision described in Theorem 3. However, one can never really check whether this condition is satisfied or not, since, for instance, $\boldsymbol{\beta}^*$ is unknown.

**Response to Reviewer # 2**   Thank you for finding our proofs sound and the paper well written. We find, however, that some other claims of your review are not fair and would like to explain our point of view.

- *This work shows that $\ell_1$-penalized Huber's M-estimator can tolerate (up to constant fraction) of outliers in the response variables (y).*

  We believe that this formulation does not reflect well the content of our paper, since the words "can tolerate" have no clear meaning in this setting. The more accurate wording would be "is minimax-rate-optimal".

- *This paper only deals with the setting where response variables (y), and has to assume that x are sub-Gaussian, and similar results were already established in (Bhatia et al., 2017).*

  Unfortunately, due to the space limits, we could not include in the paper a more detailed comparison with prior work. We will do our best for including this comparison in the revised version of the paper.

  It is not fair to compare our results to those in (Bhatia et al., 2017) for the following reasons:

  1. (Bhatia et al., 2017) consider that the number of outliers and the sparsity, or a good upper bound on it, are known, while our method does not need this information.
  2. The method in (Bhatia et al., 2017) needs the Euclidean norm of the vector $\boldsymbol{\theta}^*$, while our method does not need this information.
  3. (Bhatia et al., 2017) do not cover the case of fully adversarial contamination. They consider, for instance, that the corruption vector $\boldsymbol{\theta}^*$ is independent of the design matrix and of the noise.
  4. Finally, (Bhatia et al., 2017) do not cover the high dimensional case under the sparsity constraint. Their estimator is provably consistent only when $p/n$ goes to zero, where $p$ is the dimension of the covariates ($d$ in their paper) and $n$ is the sample size. In our result, only $s/n$ needs to go to zero, where $s$ is the number of non-zero entries of the vector $\boldsymbol{\beta}^*$. This allows us to handle the situation where $p$ is larger than $n$.

- *However, (Liu et al., 2019) can handle corruptions in both x and y, although the bound is slightly worse.*

  The words "can handle coruptions" being somewhat abstract, let us emphasize some striking differences between the results in (Liu et al., 2019) and those in our paper.

  1. (Liu et al., 2019) consider that the fraction of outliers, $\varepsilon$, and the sparsity, $s$ ($k$ in their paper) are known. They also need the norm $\|\boldsymbol{\beta}^*\|_2$ for determining the parameter $T$. None of these parameters are used by our algorithm.
  2. The constrains on $\varepsilon$ in (Liu et al., 2019, Corollary 4.1) is of the form $\varepsilon = \tilde{O}(1/\sqrt{s})$ while in our result it is of the form $\varepsilon = \tilde{O}(1)$.
  3. Most importantly, the rate of convergence in (Liu et al., 2019, Corollary 4.1), is much slower than the one in our result. Indeed, they obtain the rate $\tilde{O}(\varepsilon\sqrt{s} + \frac{\sqrt{s}}{\sqrt{n}})$ whereas our rate is $\tilde{O}(\varepsilon + \frac{\sqrt{s}}{\sqrt{n}})$.

  This being said, we agree that (Liu et al., 2019) consider a more general setting, which fully justifies the important differences mentioned above. But we feel that it is unfair to claim that their bound is "slightly worse".

We sincerely hope that all these explanations will convince the reviewer that our paper contains significant improvements of previous results. It is also paramount to stress that our results are obtained for a simple estimator that is already used by many practitioners since several decades.

**Response to Reviewer # 3**   Thank you very much for your very positive and encouraging report. We agree with all the remarks/comments/suggestions you made. Concerning your remark (6), the trace should be removed. In an initial version of the paper, Lemma 3 was stated in a more general case in which $\boldsymbol{b}$ was a matrix. This is why the trace operator was used. We apologize for this typo.

[Meta-Review · NeurIPS 2019]

Congratulations! The reviewers mostly enjoyed your work and recommended its acceptance.